

# Determination of Ice Water Content (IWC) in tropical convective clouds from X-band dual-polarization airborne radar

Cuong M. Nguyen[1], Mengistu Wolde[1] and Alexei Korolev[2]

[1]Flight Research Laboratory, National Research Council Canada, Ottawa, K1A 0R6, Canada
[2]Environment and Climate Change Canada, Toronto, J8X 4C6, Canada

*Correspondence to*: Cuong M. Nguyen (Cuong.Nguyen@nrc-cnrc.gc.ca)

**Abstract.** This paper presents a methodology for ice water content (IWC) retrieval from a dual-polarization side-looking X-band airborne radar. Measured IWC from aircraft in-situ probes is weighted by a function of the radar differential reflectivity ($Z_{dr}$) to reduce the effects of ice crystal shape and orientation on the variation of IWC - specific differential phase ($K_{dp}$) joint

distribution. A theoretical study indicates that the proposed method, which does not require a knowledge of the particle size distribution (PSD) and number density of ice crystals, is suitable for high ice water content (HIWC) regions in tropical convective clouds. Using datasets collected during the High Altitude Ice Crystal – High Ice Water Content (HAIC-HIWC) international field campaign in Cayenne, French Guiana (2015), it is shown that the proposed method improves the estimation bias by 15 % on average and reduces the root mean squared difference by 6 %, compared to the method using specific

differential phase ($K_{dp}$) alone.

## 1 Introduction

Ice water content (IWC) and its spatial distribution inside clouds are known for the significant effects they exert on the Earth's energy budget and hydrological circle (e.g. Stocker et al., 2013). Aside from its significant effect on the atmospheric processes, high ice water content ($IWC > 1\ gm^{-3}$), which is resultant from high concentration of small ice crystals in tropical mesoscale

convective systems has been linked to aircraft incidents and accidents (Lawson et al., 1998, Mason et al., 2006; Grzych and Mason, 2010; Strapp et al., 2018). Since early 1990's, over 150 engine roll-back and power-loss events have been attributed to the ingestion of ice particles produced in convective clouds (Grzych and Mason, 2010). Many studies have been undertaken to understand the details of the meteorological processes responsible for producing areas of HIWC. Equally important, methods using multi-platform observations from ground, airborne and space supplemented by weather models are being developed for

improving detection and avoidance of high IWC regions that would be potentially hazardous for aviation (Strapp et al., 2018) Conventional methods of deducing IWC from radar measurements assume a statistical relationship between the radar reflectivity factor (Z) and IWC. Such relationships are usually obtained based on IWC and Z calculated from in-situ measurements of particle size distributions (PSDs) and a size-to-mass parameterization ($m(D)$) (e.g. Heymsfield et al., 1977, Hogan et al., 2006). In recent studies (Protat et al., 2016), IWC was measured directly by bulk microphysical probes and Z



was measured from either an airborne or ground based radar. However, all of these studies show large uncertainties in the IWC-Z relationship despite the introduction of additional constraints such as air temperature (T) or the inclusion of refined $m(D)$ in the IWC calculations (Fontaine et al., 2014, Protat et al., 2016).

Lu et al. (2015) conducted an extensive simulation on both millimeter- and centimeter- wavelength radar and concluded that

the IWC-Z relationship is very sensitive to ice crystal PSDs (from one to two orders of magnitude in variability) and as such, is not recommended for IWC retrievals. Another approach employs polarimetric observations. The non-spherical geometry of ice crystals provides information on the types and habits of ice crystals (Matrosov et al., 1996, Wolde and Vali, 2001). It has been shown that the radar specific differential phase ($K_{dp}$) is less dependent on PSD, hence, is potentially useful for IWC retrieval (Vivekanandan et al., 1994; and Lu et al., 2015). Aydin and Tang (1995) suggested the possibility of combining $K_{dp}$

and differential reflectivity ratio ($Z_{dr}$) for IWC estimation for clouds composed of pristine ice crystals. However, even for the polarimetric approach, knowledge about ice crystal mass density ($\rho$) and axis ratio is needed to obtain accurate estimates of IWC. Simulation results (Lu et al., 2015) show that if only the general type of ice crystals is known, errors in IWC retrieval based on $K_{dp}$ are within 30 % of their true values. Unfortunately, the aforementioned parameters ($\rho$ and particle axis ratio) are, in general, unknown and additional assumptions are often invoked. Ryzhkov et al. (1998), for instance, took into

consideration ice crystal shapes, size-density parameterization and PSD of scatterers to reduce the uncertainty in IWC estimates.

In this paper we present a new method for assessment of IWC based on the $K_{dp}$ and $Z_{dr}$ measurements from a side-looking X-band airborne radar in tropical mesoscale convective systems (MCS). The IWC will be weighted with a function of $Z_{dr}$ to minimize the dependency of the IWC-$K_{dp}$ relationship on the particle shape and orientation, hence improve the IWC estimation

errors without knowledge of the PSD or density of the ice particles. The proposed method is examined using datasets collected during the High Altitude Ice Crystal – High Ice Water Content (HAIC-HIWC) international field campaign in Cayenne, French Guiana in May, 2015, which was carried out to enhance the knowledge of microphysical properties of high altitude ice crystal and mechanisms of their formation in deep tropical convective systems in order to address aviation safety issues related to engine icing (Strapp et al., 2018).

## 2 Background

### 2.1 Polarimetric parameters characterizing ice crystals

In conventional single-polarization Doppler radar, measured radar reflectivity, and radial velocity are used to assess cloud and precipitation spatial variability, precipitation rate and characteristic hydrometeor types. In dual-polarization radar systems, measurements are made at more than one polarization state (Bringi and Chandrasekar, 2001). Such systems can be configured

in several ways depending on the measurement goals and the choice of orthogonal polarization states. In this study, the results and discussions will be limited to the consideration of linear horizontal and vertical (H/V) polarization basis. The intrinsic



backscattering properties of the hydrometeors to the two polarization states enable the characterization of microphysical properties such as size, shape and spatial orientation of the cloud/precipitation particles in the radar resolution volume. Hence, using polarization, it is generally possible to achieve more accurate classification of hydrometeor types and estimate hydrometeor amounts such as rain fall rate. Polarimetric backscattering properties of hydrometeors depend on many factors

such as radar wavelength, radar elevation angle, particle size, shape, orientation, etc. In this section, we summarize how the differential reflectivity ($Z_{dr}$, dB) and the specific differential phase ($K_{dp}, °km^{-1}$ ) are measured by a polarimetric Doppler radar in the Rayleigh scattering regime and at low radar elevation angles.

In general, the differential reflectivity of an ensemble of n particles of size D and axis ratio r is given by (1) from Bringi and Chandrasekar (2001),

$$Z_{dr} = 10 \log_{10} \left[ \frac{|S_{hh}(r,D)|^2}{|S_{vv}(r,D)|^2} \right]$$ (1)

where $S_{hh}$ and $S_{vv}$ are the diagonal elements of the back scattering matrices.

The specific differential phase is defined as,

$$K_{dp} = \frac{2\pi n}{k} Re\left[ \vec{f}_{hh}(r,D) - \vec{f}_{vv}(r,D) \right]$$ (2)

where $k$ is wavenumber, $Re[\ ]$ stands for the real part of a complex number and $\vec{f}_{hh}, \vec{f}_{vv}$ are the forward scattering amplitudes

at horizontal and vertical polarization, respectively. Equation (2) shows that $K_{dp}$ is proportional to n. Consequently, for a large number of small particles with the axis ratio close unity ($r \approx 1$), $Z_{dr} \rightarrow 0\ dB$, whereas $K_{dp}$ can be large, even if the second term in Eq. (2) becomes small.

In a simple form of the calculations of $Z_{dr}$ and $K_{dp}$ of ice crystals, it is customary to approximate columns as homogeneous prolate spheroids and plates as homogeneous oblate spheroids. In the case of side incidence, the elevation angle is assumed to

be close to zero and there is no (or very small) canting in the vertical plane. In the absence of wind shear and turbulence, and assuming a perfectly aligned spheroid model, $Z_{dr}$ and $K_{dp}$ can be expressed as functions of ice particle size, axis ratio and the relative permittivity of the particle ($\boldsymbol{\varepsilon}$) (Bringi and Chandrasekar, 2001 ). For example, for oblate spheroid ice particles with a particle size distribution, $N(D)$,

$$|S_{hh}(r,D)| \approx \frac{k^2}{4\pi} \frac{V(D)|\varepsilon-1|}{\left[1 + \frac{1}{2}(1-\lambda_o)|\varepsilon-1|\right]}$$ (3)

$$|S_{vv}(r,D)| \approx \frac{k^2}{4\pi} \frac{V(D)|\varepsilon-1|}{[1 + \lambda_o|\varepsilon-1|]}$$ (4)

$$Z_{hh,vv} = \frac{\lambda^4}{\pi^5 K_p^2} \int 4\pi \left|S_{hh,vv}\right|^2 N(D)dD$$ (5)

$$K_{dp} = \frac{k}{2} \int \underbrace{\left[ \frac{|\varepsilon-1|}{\left[1 + \frac{1}{2}(1-\lambda_o)|\varepsilon-1|\right]} - \frac{|\varepsilon-1|}{[1 + \lambda_o|\varepsilon-1|]} \right]}_{\alpha} V(D)N(D)dD$$ (6)





where $\sigma_{hh,vv}$ is the radar cross section, $K_p$ is dielectric factor (for ice, we are using $K_p^2 = 0.177$) and $V(D)$ is the particle volume. $\lambda_o$ is the depolarizing factor which is only a function of the axis ratio $r = b/a$ (for oblate particles, a is the semi-major axis length and b is the semi-minor axis length ($a > b$)).

$$\lambda_o = \lambda(oblate) = \frac{1+f^2}{f^2}\left(1 - \frac{1}{f}\tan^{-1}f\right); \qquad\qquad f^2 = \frac{1}{r^2} - 1 \qquad\qquad (7)$$

5     A similar equation for $K_{dp}$ can also be derived for prolate spheroid ice particles with symmetry axis parallel to the horizontal plane (Hogan et al., 2006).

On other hand, the IWC can be defined in terms of the size distribution,

$$IWC = \int \rho(D)V(D)N(D)dD \qquad\qquad\qquad (8)$$

where $\rho(D)$ is the mass density of ice crystals with size D.

## 2.2 Polarimetric methods for IWC retrieval

An inspection of Eqs. (3) and (4) suggests that for small ice crystal particles, the radar cross section ($\sigma_{hh,vv} = 4\pi|S_{hh,vv}|^2$) is roughly proportional to the square of the ice particle mass ($\rho_i^2(D)V(D)^2$), a conclusion also confirmed by results from simulated data (Lu et al., 2015). In addition, according to Lu et. al. (2015), for particles of size comparable or larger than the radar wavelength, there is no clear relationship between the radar cross section and ice particle mass due to the Mie resonance effects. In either case, $\sigma_{hh,vv}$ is not directly proportional to the particle mass. Hence, the $Z - IWC$ relationship depends strongly on the particle size distribution and the radar frequency. Consequently, using Z only to estimate IWC without knowledge of the PSD can lead to errors as large as one order of magnitude. On the other hand, Eq. (6) indicates that if the terms in square brackets ($\alpha$), are proportional to the ice density ($\rho(D)$), then the $K_{dp} - IWC$ relationship is independent of PSD. The proportionality constant depends on several factors such as the ice crystal type, orientation and the measurement elevation angle. It is shown that the variability of this proportionality constant significantly increases at large elevation angles (Lu et al., 2015). Further, when the exact ice crystal type is known, averaged relative error in the estimated IWC using $K_{dp}$ can be as small as 10 %, regardless of whether PSD is known or not. If the ice crystal types are unknown but can be generally categorized, the errors can be higher, but mostly less than 30 %. These numbers were averaged from elevations in the interval $[0° - 70°]$. If IWC is estimated using $K_{dp}$ at small elevation angles (less than 10°) such as from a side looking antenna, we would expect better results.

It is noted that $K_{dp}$ is sensitive to the shape and the orientation of the ice crystal (Eq. (7)) while the IWC is not. Consequently, in the case of spatial variability of ice crystal shapes and orientations, the IWC estimation based on $K_{dp}$ may be biased. To mitigate this problem, the dependency of IWC estimates on the ice particles' shapes must be removed. One way to do this is to weight the measured IWC by a function of ice crystal shapes and orientations before applying a linear regression model to the $K_{dp} - IWC$ relationship. In a simple approach, the weighting function can be in a form of $Z_{DR}^a$ ($Z_{DR}$ is the linear version of $Z_{dr}$) as suggested in Aydin and Tang (1997) (derived from their approximation $IWC \approx K_{dp}^a Z_{DR}^b$). Proceeding more



rigorously, Ryzhkov et al. (1998) demonstrated that both $K_{dp}$ and difference reflectivity $Z_{DP}$ ($Z_{DP} = Z_H - Z_V$) are dependent on the particle shape, whereas their ratio is mainly determined by the mass of the particle. This approximation is probably suitable for small ice crystal (median mass diameter (MMD) less than 1 mm). More specifically,

$$Z_H \approx \int M(D)^2 N(D) dD \tag{9}$$

$$IWC \approx \int M(D) N(D) dD \tag{10}$$

$$\frac{Z_{DP}}{K_{dp}} \approx M \tag{11}$$

where and $M = \rho V$ is the ice particle mass. Using Eqs. (9) – (11) along with an assumption of exponential form of particle size distribution $N(D)$ one can easily derive a closed form of IWC as a function of $K_{dp}$ and $Z_{DR}$. Alternatively, if the particle mass variation is small within the radar volume, it is straightforward to obtain the approximation $(1 - Z_{DR}^{-1})IWC \approx K_{dp}$ without any assumption on the form of $N(D)$. This assumption might not be suitable for all the types of ice clouds but is might be suitable for HIWC regions which are often composed of high concentration of small ice particles (Leroy et al., 2016).

At $Z_{DR} \approx 1$ (or $Z_{dr} \approx 0 \ dB$), the weighting function $(1 - Z_{DR}^{-1})$ is close to zero; and hence, it can introduce large errors in the estimates. Therefore, there should be a certain threshold for $Z_{dr}$ to determine how the weighting function would be calculated. In detail, if $\mathbf{Z_{dr}}$ is less than a threshold, the weighting function $(1 - Z_{DR}^{-1})$ is replaced by $(1 - Z_{DR-threshold}^{-1})$. In this paper, we use $Z_{dr-threshold} = 0.6$ dB (or $Z_{DR-threshold} = 1.15$) as proposed by Ryzhkov et al. (1998) for "cold" storms (below $-5 \ ^oC$) (see the measurement temperature ranges in the next section).

In summary, there are two polarimetric methods for IWC retrieval, which will be investigated and compared in this paper. They are expressed as,

$$IWC = a_1 K_{dp} + b_1 \tag{12}$$

$$(1 - Z_{DR}^{-1})IWC = a_2 K_{dp} + b_2 \tag{13}$$

where model parameters $(a_i, b_i)$ will be estimated from measured data.

## 3 Airborne measurements

During the Cayenne HAIC-HIWC project, the NRC Convair-580 conducted fourteen research flights in both continental and oceanic mesoscale convective systems with high IWC. For this campaign, the Convair aircraft was instrumented by the NRC and Environment and Climate Change Canada with an array of in-situ cloud microphysics probes, atmospheric sensors and the NRC Airborne W- and X-band (NAWX) Doppler dual-polarization radars (Wolde and Pazmany, 2005). The unique quasi-collocated in-situ and radar data collected during the HAIC-HIWC mission provided a means for developing techniques for detection and estimation of high IWC that could be adopted in operational airborne weather radars.



### 3.1 Airborne radar data

In this study, dual-polarization radar data from the NRC airborne X-band radar (NAX) (Fig. 1) side looking antenna is used. In the Cayenne project, the radar complex I and Q samples are processed to powers and complex pulse pair products according to the radar parameter specifications table and the products are recorded in binary format. Due to the size of the aircraft radar

radome, the NAX dual-polarization parabolic side antenna is relatively small (26 ''), hence, exhibits some limitations in terms of sidelobe performance. The antenna OMT/feedhorn combination is relatively large compared to the parabolic dish. The large feed structure creates some significant sidelobes at ±90 ° planes. As a result, when the sidelobes intercept targets with strong returns below the aircraft, such as the earth surface or a storm melting layer, significant returns from the sidelobes will contaminate signals coming via the antenna's main lobe. In most situations, the effect is more prominent at a range equal to or

greater than the distance where the antenna sidelobes hit the ground. At regions where signals are contaminated by ground clutter via the sidelobes, the data is intermittent and exhibits large biases. Unfortunately, with the pulse pair data from the Cayenne campaign, methods to separate clutter from the precipitation signals are limited. To overcome this issue, a method is developed to detect regions with strong clutter contamination based on signal correlations between the nadir and zenith returns. If the correlation coefficient exceeds a pre-defined threshold, the corresponding side data in those regions is discarded. If the

width of the discarded data region is relatively small (less than 300 m in radar range) it will be filled through interpolation. In addition, due to the limitation of the radar hardware, the measurements of dual-polarization parameters are not useable below a range of 1000 m from the aircraft, but reflectivity can be measured accurately from 450 m. This is not an ideal condition, when the in-situ data and the radar data are not spatially coincident. However, in most scenarios the advantage of having very fine radar sampling volumes with high order of accuracy in time synchronization between in-situ probes overcomes the location

offset. At large distances from cloud boundaries the microphysics properties of ice clouds with a good accuracy can be considered spatially quasi-uniform at scales of the order of few hundred meters. This is specifically relevant to the measurements in MCSs during the HAIC-HIWC project.





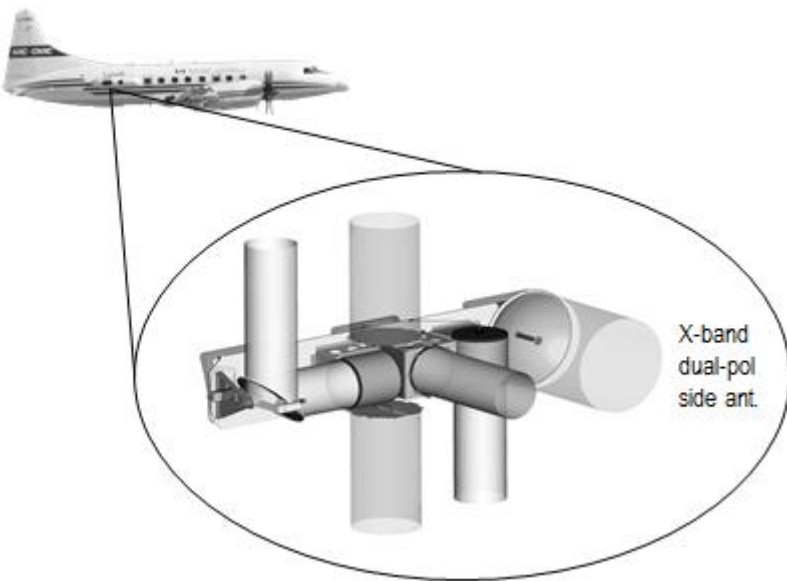

**Figure 1: The NRC Convair-580 and the dual-polarization side-looking X-band radar.**

## 3.2 In-situ data

For the project, the NRC Convair-580 was equipped with state-of-the-art in-situ sensors for measurements of aircraft and

atmospheric state parameters and cloud microphysics. There were multiple sensors to measure bulk liquid water content (LWC) and total water content (TWC), hydrometeor size distribution ranging from small cloud drops to large precipitation particles. Detailed list of the Convair in-situ sensors used during the Cayenne HIWC-HIWC project are provided in Wolde et. al. (2016). Here we will briefly describe the in-situ microphysical sensors used in correlating the airborne radar measurements with regions of HIWC. TWC was measured by a hot-wire Nevzorov probe (Korolev et al., 1998) and an Isokinetic probe

(IKP2) that was specifically designed to measure very high TWC (Davison et al., 2016). The IWC from the Nevzorov probe was obtained from the TWC and LWC as described in Korolev and Strapp (2002). Due to the bouncing effect the Nevzorov TWC sensor underestimates condensed water in ice clouds by approximately two times, and therefore, this instrumentation is not fully suitable for high IWC measurements. As shown in Korolev et al. (2018) in the MCSs studied during the Cayenne HAIC-HIWC project, the fraction of mixed phase clouds at -15 ℃<T<-5 ℃ did not exceed 4.6 %, and that in most mixed

phase cloud regions LWC<<IWC. Therefore, with a good accuracy for this specific data set it can be assumed that TWC=IWC. This finding significantly simplifies the processing and interpretation of cloud microphysical measurements. Alternatively, IWC was estimated from the measured PSDs ($IWC_{PSD}$) with the D-M parameterization was tuned using IKP2 measurements. In the Cayenne Convair datasets, IWCs calculated from PSDs and measured by IKP2 agreed quite well and the difference between them in the HIWC regions on average did not exceed 15 %. Because the IKP2 data was not available in all flights,

estimated IWC from PSDs has been used in this work. Additionally, mean mass diameter (MMD) was also used to characterize



the microphysical properties of the high IWC regions and interpret X-band radar measurements. MMD was calculated from composite particle size distributions measured by SPEC 2D-S and DMT PIP 2D imaging probes.

## 4 $K_{dp}$ estimation algorithm for X-band airborne weather radar

The radar specific differential phase ($K_{dp}$) is defined as the slope of the range profile of the differential propagation phase

shift $\Phi_{dp}$ between horizontal and vertical polarization states (Bringi and Chandrasekar, 2001). The measured differential phase shift between the two signals at the H and V polarizations, $\Psi_{dp}$, contains both $\Phi_{dp}$ and differential backscatter phase shift $\delta_{dp}$. If $\delta_{dp}$ is relatively constant or negligible, the profile of $\Psi_{dp}$ can be used to estimate $K_{dp}$. The differential propagation phase shift $\Phi_{dp}$ is a continuous range function, whereas the estimated $\Psi_{dp}$ usually exhibits discontinuities due to phase wrapping, statistical fluctuations in estimation and the gate-to-gate variation of $\delta_{dp}$. Because the statistical fluctuations in the estimates

of $\Psi_{dp}$ will be magnified during the differentiation, resulting in a large variance of the $K_{dp}$ estimates, the following considerations need to be addressed in the $K_{dp}$ estimation algorithm.

- Phase unfolding: phase wrapping occurs when the total $\Phi_{dp}$ accumulation exceeds the unambiguous ranges. This depends on the system differential phase $\Phi_{dp}(0)$ and the cumulative phase due to the medium. The NAX radar operates in the simultaneous transmission mode (VHS) and the unambiguous range is 360 °. The system differential

phase $\Phi_{dp}(0)$ of NAX is about 64 °. For the Cayenne data set, no observations have been made when the phase was folded.

- $\delta_{dp}$ "bump": it seems that $\delta_{dp}$ was negligible in the HIWC environment in the Cayenne campaign. We did not observe the presence of significant changes in $\delta_{dp}$ over a short range.

- Range filtering: in this work, the range scale was set at 500m, thus, the fluctuations at scales smaller than 500 m will

be suppressed.

Once the phase data is quality controlled, filtered and decimated to match the temporal resolution of the in-situ data, a heuristic algorithm similar to one reported in Rotemberg (1999) is applied to the data to extract $\Psi_{dp}$ smooth trend and then $K_{dp}$ is computed from it. This approach does not require an assumption of $\Phi_{dp}$ being a monotonically increasing function as it is in some other existing $K_{dp}$ retrieval algorithms (Wang and Chandrasekar, 2009); therefore, it would also work well with negative

$K_{dp}$ which possibly exhibits in ice clouds. Our preliminary analysis shows that the algorithm can provide estimates with standard deviation no greater than $1\ ^okm^{-1}$. The NRC $K_{dp}$ estimation algorithm is summarized in the flowchart below.

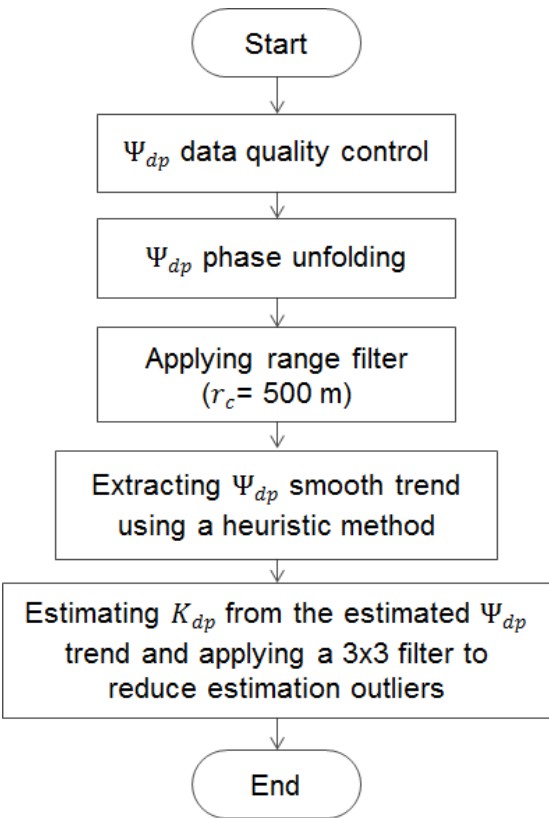

**Figure 2: The NRC $K_{dp}$ estimation algorithm for X-band radar diagram.**

## 5 Results

In this section, preliminary results illustrating the performance of the proposed polarimetric algorithms are presented. Besides

5 the polarimetric method, we also include results from the conventional $IWC - Z$ relations for comparison. Because the

histogram of static temperature (not shown) indicated a bimodal distribution with two centers at around -5 ℃ and -10 ℃, two

$IWC - Z$ relations at $T = -5 \,^oC$ ($IWC = 0.257Z^{0.391}$) and at $T = -10^oC$ ($IWC = 0.253Z^{0.596}$) were obtained by fitting

power-law curves to scatter plots of all the data points at those two temperature levels (Wolde et al., 2016).

### 5.1 Case study I: May 26 flight

10 In this case, a 20-minute segment of the flight on May 26, 2015 is selected. Figure 3a shows IR satellite imagery obtained

during the flight where the aircraft's flight track is shown in different colors, which represent the aircraft's location at different

time segments. The reflectivity field from the NAX side antenna is shown in Fig. 3b. The selected period begins at a point

(white segment) when the aircraft started to enter the convective core of the storm within proximity of the cells with the lowest

cloud top brightness temperature. The brightness temperature was increasing toward the end of the segment (cyan segment).





In addition to the radar data, IWC and MMD time series from particle probes and Nevzorov probe are shown in Fig. 4. The radar estimates have been decimated to match with the temporal resolution of the in-situ data.

The aircraft sampled two regions: a convective region before 11:23 UTC and a stratiform region after 11:25 UTC (Fig. 3b), with $IWC$ in both regions was mostly higher than $1.5 \, gm^{-3}$ (Fig. 4a). It is worth noticing that the reflectivity measurements along the flight path was $\sim 20 \, dBZ$ and the MMD values were relative small at most locations (Fig 4a). This means the ensemble of particles might be presented by a mixture of large aggregates and small pristine ice particle with dominating reflectivity of small ice crystals. Initial observations from Fig. 4 include: (1) $K_{dp}$, in general, is highly correlated with IWC; (2) regions with larger MMD exhibits deceasing $\rho_{hv}$ and increasing $Z_{dr}$. Measurements of the latter are reflectivity weighted and biased towards the $Z_{dr}$ of larger particles while $K_{dp}$ is only sensitive to the small oriented crystals (as be seen between 11:24 to 11:27 UTC). In Fig. 5, $Z_{dr}$, $\rho_{hv}$ and IWC are expressed as functions of $K_{dp}$. In this case, there is a break point at $K_{dp} \approx 1^o$ (and $Z_{DR} \sim 1.15$) where $Z_{dr}$ started increasing and $\rho_{hv}$ deceased with respect to $K_{dp}$. At $K_{dp} < 1^o$, $Z_{dr}$ was mainly flat and IWC linearly increased with respect to $K_{dp}$ (Fig. 5b). This suggests the pristine ice crystals' axis ratio might be fairly constant but the particle number density increased resulting in an enhancement in both $K_{dp}$ and IWC (shown by a linear IWC-$K_{dp}$ relationship). From $K_{dp} > 1^o$, $Z_{dr}$ increment with respect to $K_{dp}$ was greater, but IWC growth did not follow the same degree as in the previous segment. If a linear IWC-$K_{dp}$ relationship derived from the first segment ($K_{dp} < 1^o$) is applied the second portion, IWC will be overestimated. Many factors could contribute to this circumstance such as changes in ice crystals' size, shape, orientation (e.g. particle with higher axis ratio aligned in the horizontal plane) or particle's density at the second segment. It is not easy to identify the exact reasons of this process; nevertheless, the main objective of this work is to determine if additional parameters such as $Z_{dr}$ and $\rho_{hv}$ can be used to mitigate this dependency and improve estimation of IWC. In Fig. 5c, the modified version of IWC (Eq. (13)) is shown in solid blue line. Additionally, linear fitting lines computed from the two datasets, measured IWC ($IWC_{meas}$) and modified IWC ($IWC_{mod}$) are also superimposed. The $R^2$ goodness of fit parameter indicates that a linear regression fits the modified IWC better in comparison to the original IWC.



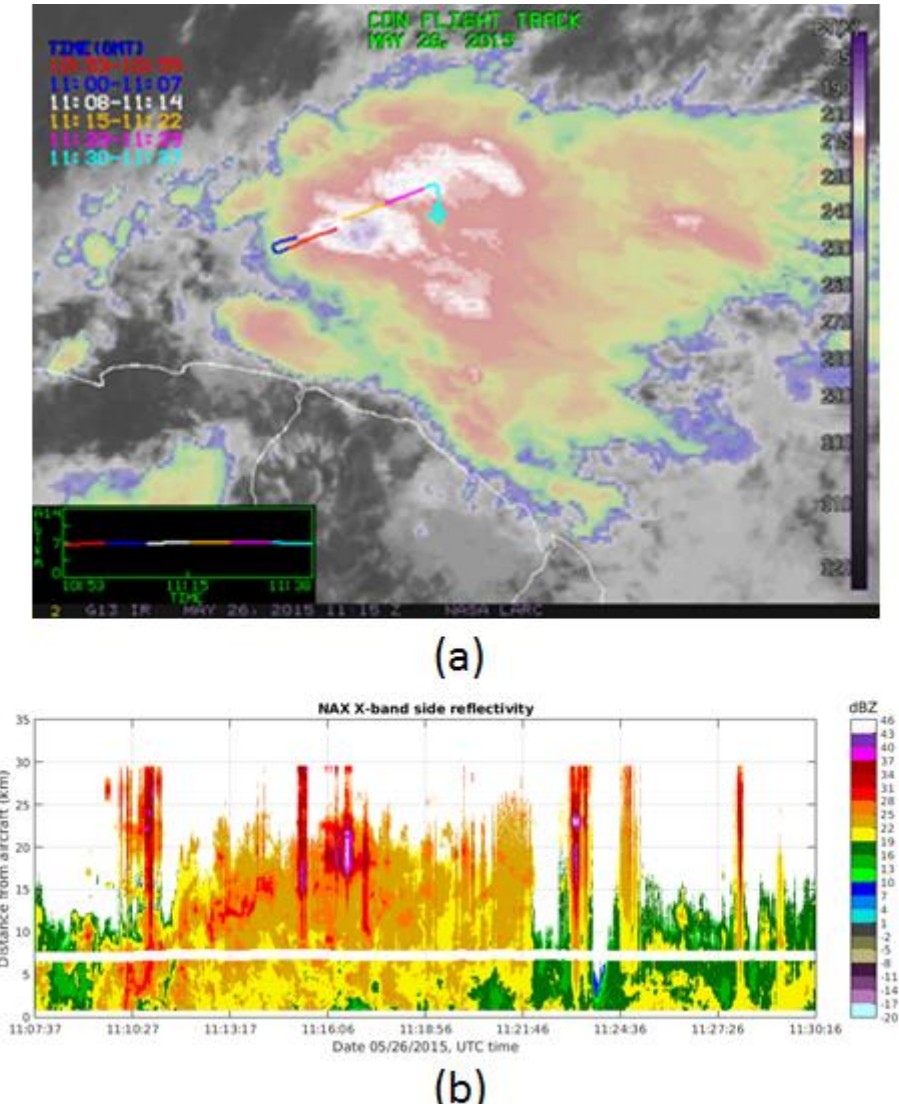

**Figure 3: Top panel shows IR GOES-13 image with the overlaid segments of the Convair580 flight track on May 26, 2015. Different time segments of the flight track are shown by different colors. Bottom panel is shows X-band side reflectivity from a period of [11:07 - 11:30] UTC corresponding to white, yellow and purple segments in the top panel.**



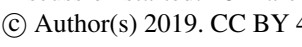

**Figure 4: Time series of (a) IWC, MMD, (b) $K_{dp}$, $Z_{dr}$ , (c) $\rho_{hv}$ and $Z_H$ for May 26 Convair-580 flight.**



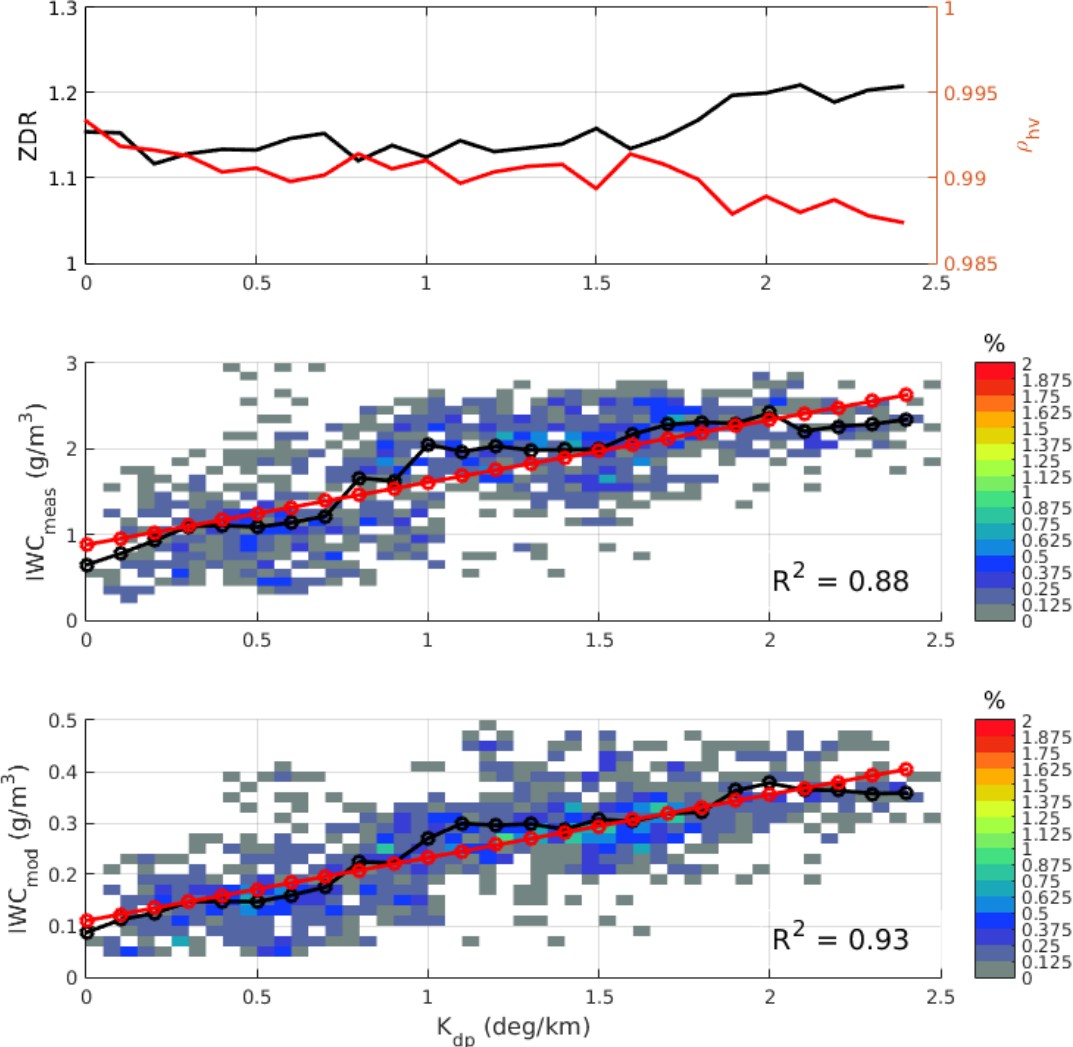

**Figure 5:** $Z_{dr}$ and $\rho_{hv}$ (a), $IWC_{meas}$ (b) and $IWC_{mod}$ (c) as functions of $K_{dp}$. Mean values and scatter plots are computed from data points in each $K_{dp}$ bin of $0.1^o$ and $0.05^o$ respectively.

To gauge the performance of the polarimetric methods, results from the conventional $IWC - Z$ estimator are also included. In

5   Fig. 6, the measured IWC along the Convair's flight path is depicted in blue, IWC-Z result is shown in green and IWC estimates using polarimetric methods are shown in blue and red for $K_{dp}$-only and $(K_{dp}, Z_{DR})$ algorithms, respectively. One can observe that the two polarimetric methods agree well with measured $IWC$ from the Nevzorov probe while the IWC estimates from radar reflectivity exhibit biases as large as one order of magnitude. The large errors in the $IWC - Z$ estimator are due to the presence of mixtures of large aggregates and small ice crystal regions as indicated in the PIP images (not shown) in clouds.





Large aggregates have a dominant contribution into the radar reflectivity, which explains the positive biases of the $IWC - Z$ estimates. In contrast, $K_{dp}$ is more sensitive to the oriented small ice crystals ($K_{dp}$ of large aggregates is near zero). It follows that estimators utilizing $K_{dp}$ information would be able to overcome the effects of large aggregates in radar volumes. It is worth noting that the two algorithms capture well the IWC variation at the end of the segment. If the in-situ measurements are considered as the ground truth, the estimations errors are computed and shown in Fig. 6b. On average, both polarimetric estimators provide unbiased results, whereas the root mean squared differences (hereinafter referred to as the rms differences) are 0.48 $gm^{-3}$ and 0.45 $gm^{-3}$ for the $K_{dp}$ alone and ($K_{dp}, Z_{DR}$) methods, respectively. The correlation coefficients between the in-situ and estimated IWCs are 0.64 and 0.70 for the two methods. In this case study, the inclusion of $Z_{dr}$ improves the accuracy of the IWC estimates.



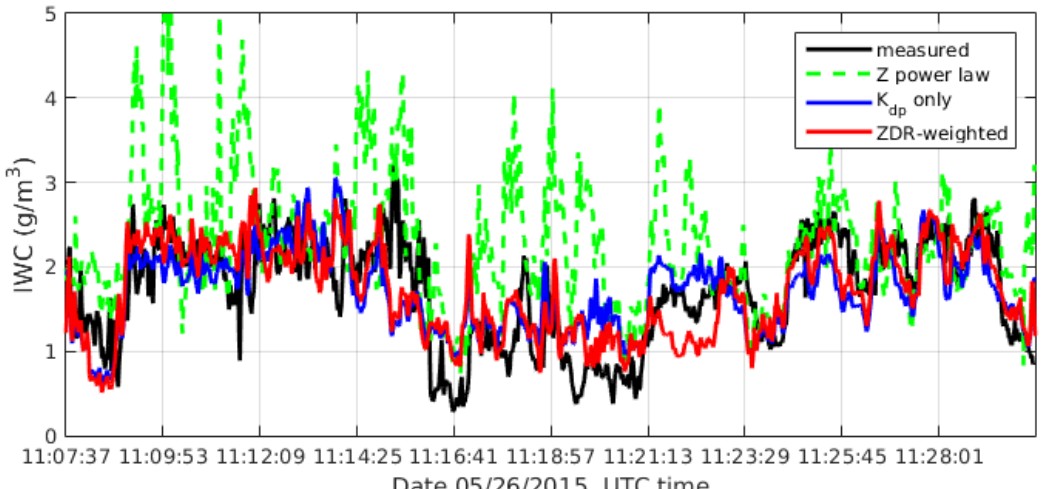

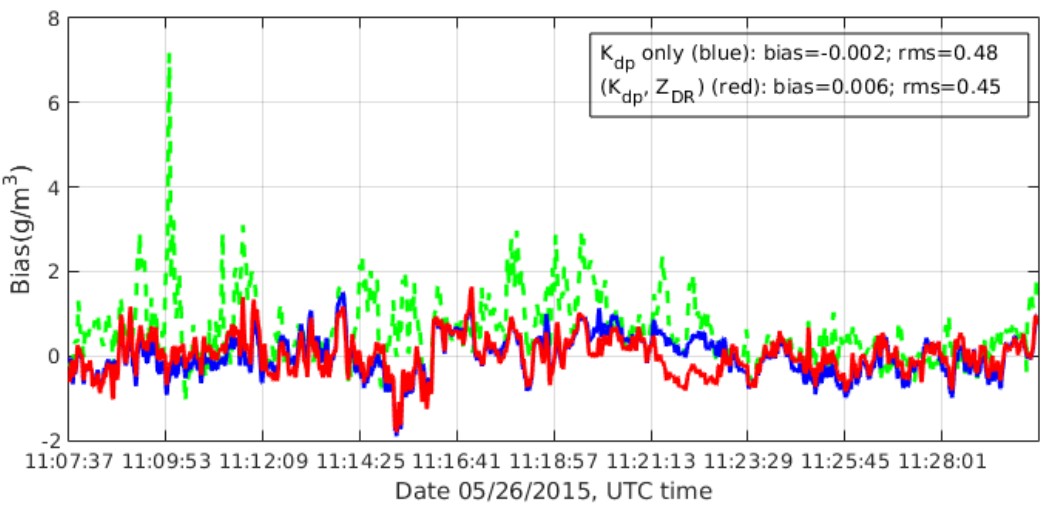

**Figure 6: Top panel shows measured IWC from the Nevzorov (black line), estimated IWC using reflectivity (dash green line), $K_{dp}$ alone (blue line) and ($K_{dp}$, $Z_{DR}$) combination (red line) for the May 26 case. Bottom panel shows estimation errors for the three estimators. Average biases for IWC($K_{dp}$) and IWC($K_{dp}$, $Z_{DR}$) are -0.002 $gm^{-3}$ and 0.006 $gm^{-3}$ and rms differences are 0.48 $gm^{-3}$ and 0.45 $gm^{-3}$ for the two algorithms, correspondingly.**

**5.2 Case study II: May 23 flight**

For this case, a segment consisting of a region of high *IWC* of very high density of small ice particles and a region of mixture of moderately large aggregates and pristine ice crystals was analysed. This affords an excellent example to gauge the

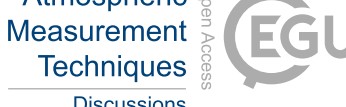



performance of the algorithms. In Fig. 7a the selected segment is displayed in purple. The radar reflectivity field from the side antenna is shown in Fig. 7b.

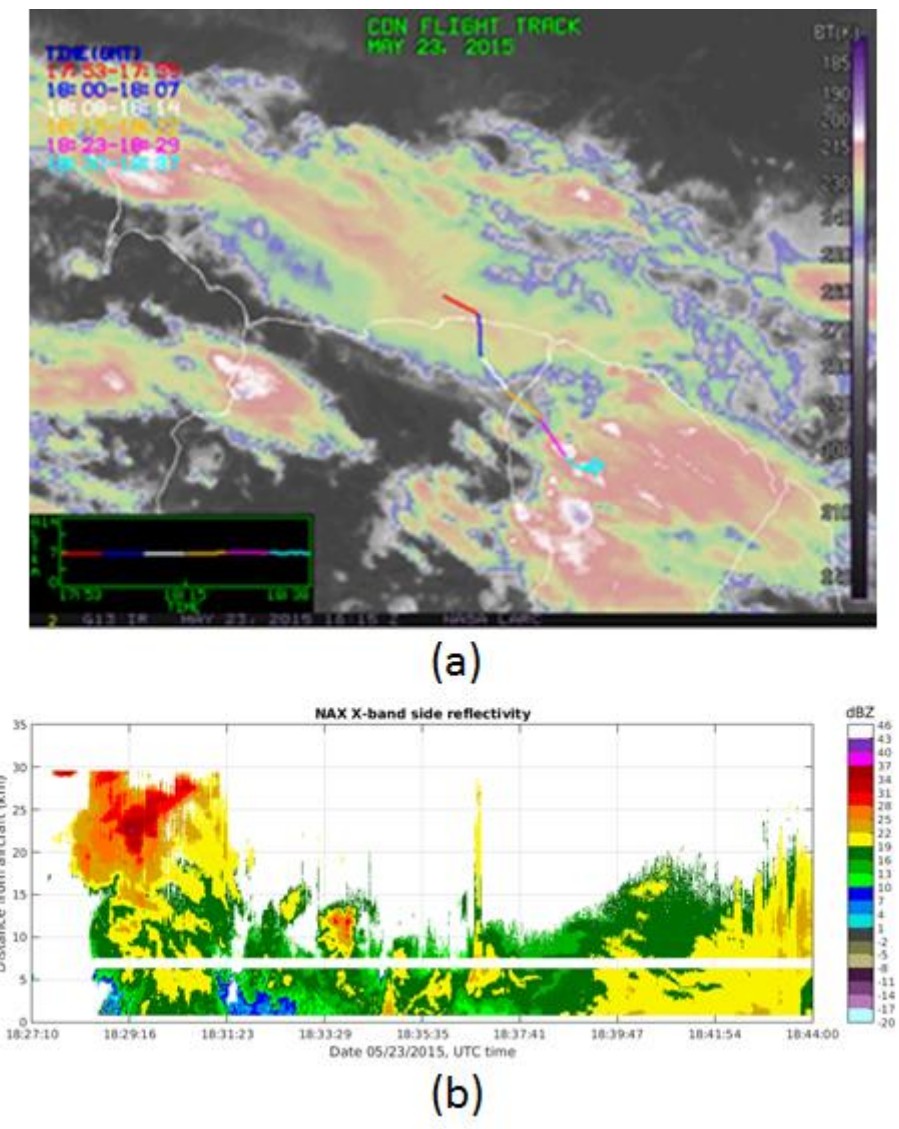

**Figure 7: Similar to Fig. 3 but for the May 23 case.**

5    In addition to the radar data, $IWC_{PSD}$ and MMD time series particle probe are shown in Fig. 8. The aircraft sampled two small cores where $IWC$ was higher than 1 $gm^{-3}$ (~18:30 UTC, and ~18:34 UTC). In these high IWC cores, the MMD was in the 400 μm range. In contrast, for the flight segment between 18:36-18:44 UTC, when the temperature was higher, the aircraft



sampled a mixture of large aggregates with sizes exceeding 6 mm, and small ice particles (Fig. 9), where the IWC was less than 0.5 $gm^{-3}$.

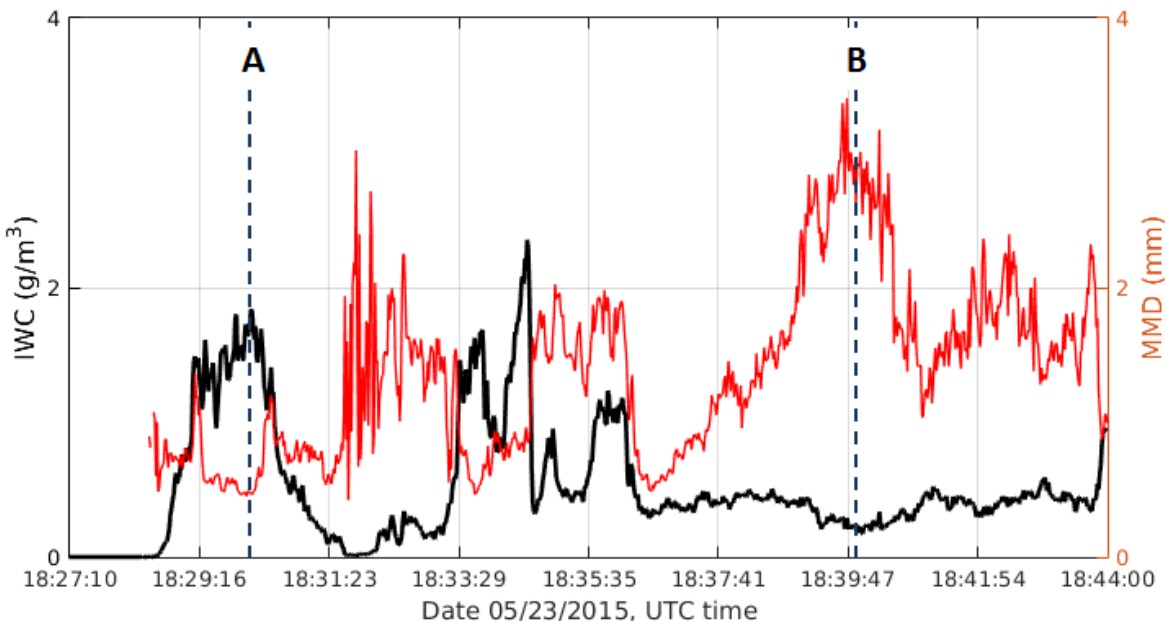

**Figure 8: IWC and MMD time series for the May 23 case.**

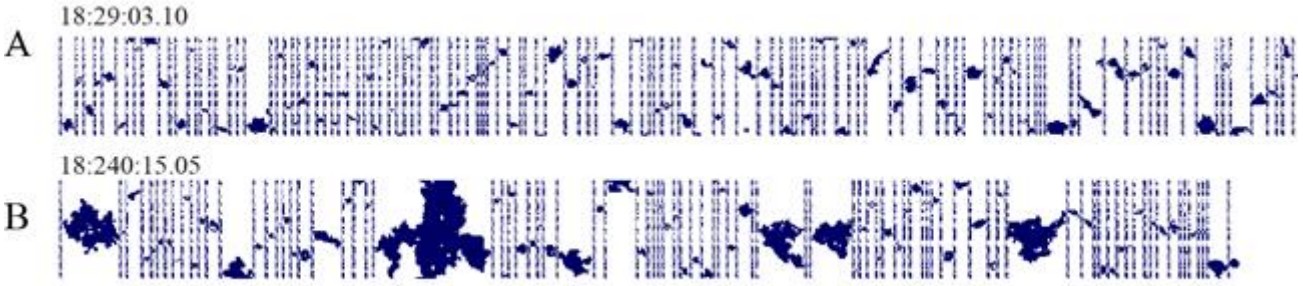

**Figure 9: Sample of 2D imagery from the DMT PIP probe at two time stamps as in Fig. 9. The width of the PIP image strips is 6.4 mm.**

The IWC estimates from the methods are plotted in Fig. 11a. In the region dominated by small particles (before 18:33:29 UTC), results from the three estimators agree quite well with $IWC_{PSD}$. There are small biases in the outcomes of the two polarimetric algorithms at the two HIWC peaks. These biases can be attributed to the errors of fitting linear regression models to the data and/or the difference in the sampling locations of the radar and the in-situ data (section 3.1). In the region after 18:38:01 UTC, the $IWC-Z$ results show very large errors due to the presence of mixtures of aggregates and ice crystals



(shown in the PIP imagery in Fig. 9 and in high resolution 2DS particle imagery (Fig. 10) in the clouds. The large aggregates dominate the measurements of radar reflectivity, which explains the positive biases of the $IWC - Z$ estimates. The errors for this case are as large as 300 % in most estimates. In contrast, both the polarimetric methods provide much better results compared to the conventional IWC-Z method. They capture well the variation of IWC at smaller scales (around 18:33:58 UTC)

and larger scales (after 18:36:14 UTC). This again confirms that these algorithms are robust to the variation of ice crystal type, shape and distribution. The rms differences and correlation coefficients for $K_{dp}$-only and ($K_{dp}$, $Z_{DR}$ ) methods are (0.52 $gm^{-3}$, 0.62) and (0.45 $gm^{-3}$, 0.67), respectively. The combination of $K_{dp}$ and $Z_{DR}$ provides better results which can be seen at the edges of the second IWC peak (indicated by ellipses) in Fig. 11a. At those regions, MMD (Fig. 8) and $Z_{dr}$ (not shown) values are large. This may be an indication of ice crystals with high axis ratio aligned in the horizontal plane. When this happens, the

algorithm based on $K_{dp}$ alone will over-estimate IWC. On the other hand, the modified IWC is less sensitive to the particles' shape and orientation, thus, estimates based on it yield better results. When large particles dominated the volume (after 18:36:14 UTC) ($Z_{dr} \rightarrow 0\ dB$) then the $\left(K_{dp}, Z_{DR}\right)$ estimator provides no advantage over the $K_{dp}$-only estimator.

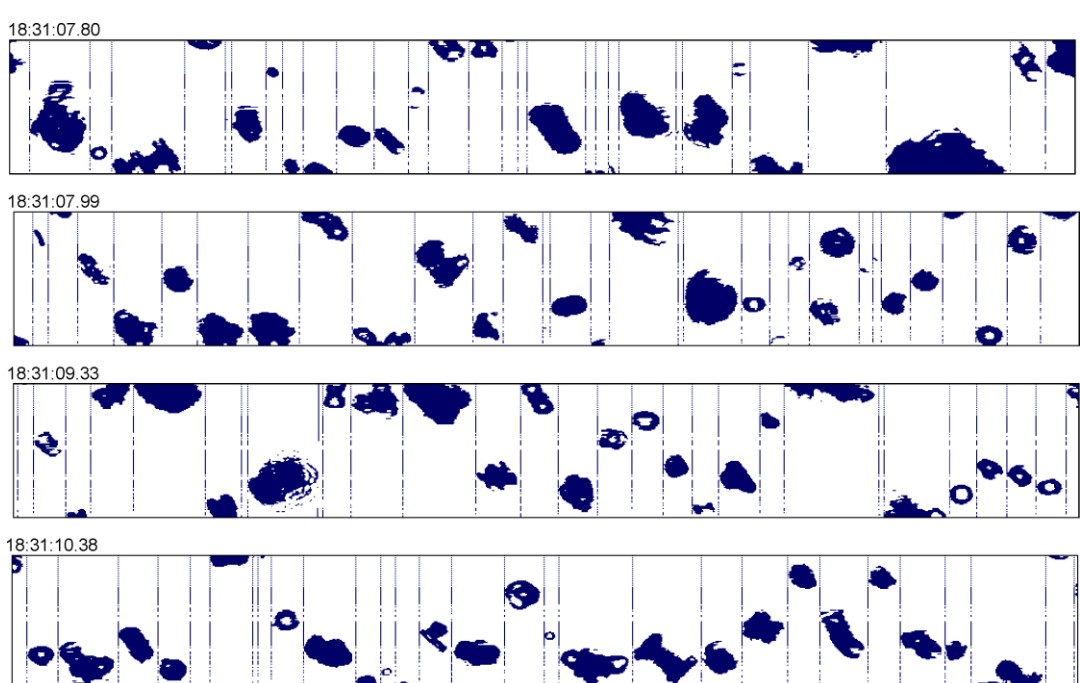

**Figure 10:  Images of ice particles sampled by the SPEC 2DS probe on the flight on 23 May 2015 The width of the vertical strip is 1.28 mm.**



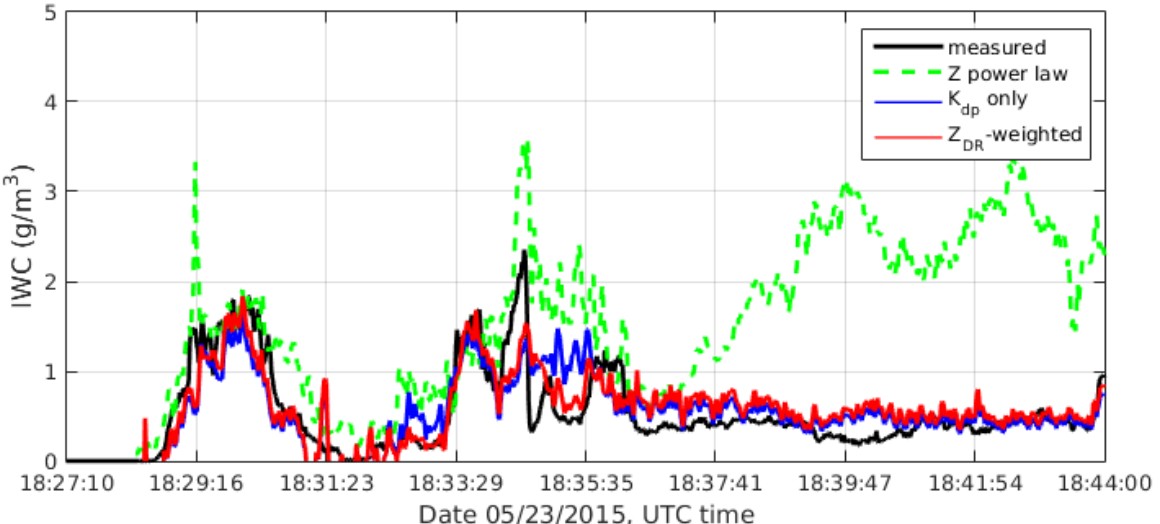

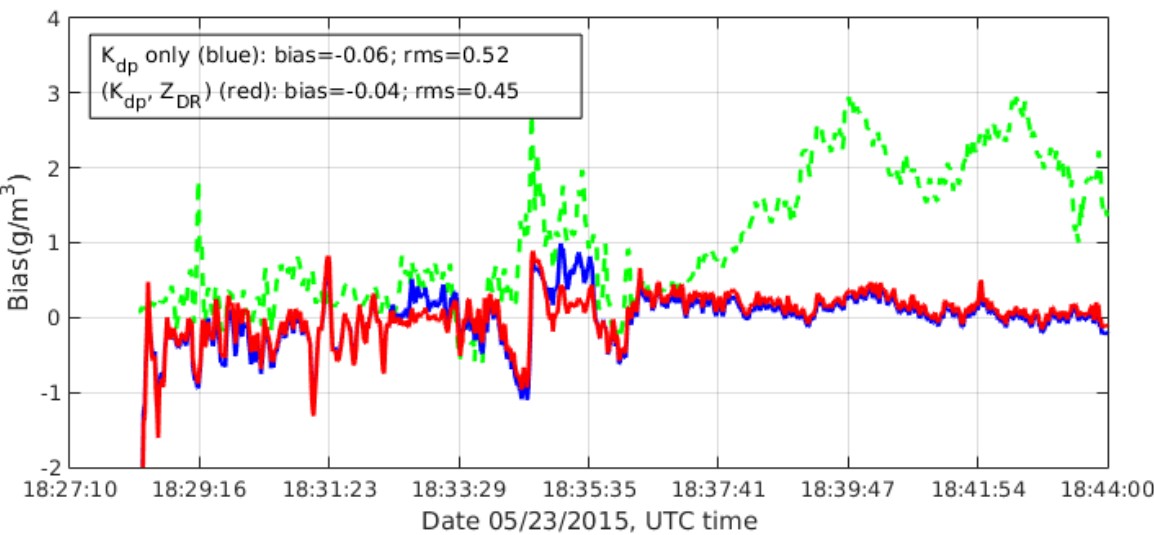

**Figure 11: Similar to Fig. 7 but for the May 23 case.**

## 6 Experimental evaluation

In the previous section, two case studies were analyzed in detail. In both cases, results from the polarimetric methods show a

5   much better agreement with in-situ measurements compared to the IWC estimates from the radar reflectivity factor, especially



when larger particles dominate the radar volume. In addition, applying a function of $Z_{DR}$ to IWC before fitting a linear regression model to the data improves the estimation accuracy and correlation. In this section, more data from different flights collected during the mission were analysed and summarized. Out of total 14 campaign flights, there were seven flights with good data quality (radar and in-situ) and with applicable number of high IWC data points and data from those flights were

used this analysis.  In Fig. 12, IWCs and modified IWCs are expressed as functions of $K_{dp}$ for the selected flights. For most cases, the modified version of IWC is better replicated by a simple linear regression model. It can be seen that these linear relationships are well approximated up to $K_{dp} = 2^{o}$. At larger $K_{dp}$, IWC saturates at 2.5 $gm^{-3}$ and the IWC-$K_{dp}$ relationship departs from the linear trend. Due to the limited amount of data of large measured $K_{dp}$ and IWC, identifying the major reasons for this saturation is not attempted. In these scenarios, applying a more sophisticated method (such as a parametric model) will

likely reduce errors at high $K_{dp}$ but this is beyond the scope of this paper. Here, a simple linear regression model (based on the approximation in Eq. (13)) is used and errors are computed from all data points.

In Fig. 12, it is also worth noting that the scattering of the modified IWC-$K_{dp}$ curves is significantly narrower compared to that of the original IWC-$K_{dp}$ curves. The scattering in IWC-$K_{dp}$ relationship can be attributed to the properties of ice crystals and the medium's state. In other words, when the dependency of IWC-$K_{dp}$ relationship on ice crystal shape and orientation

was removed (or partially removed), the scattering of IWC-$K_{dp}$ should be tighter. This is a very important outcome which helps to reduce estimation errors when a single estimator is used for all the cases. Results for IWC estimates are shown in Table 1 for the two polarimetric methods only. In each row, statistical error analysis is shown for each flight with the optimal fitting model derived from data of that flight. The last row displays results computed from all selected data of 17699 points. In all cases, improvement in IWC estimation when $Z_{dr}$ information is utilized in the algorithm is clear. For all data, the rms

difference reduces from 0.52 $gm^{-3}$ to 0.49 $gm^{-3}$ and correlation coefficient increases from 0.69 to 0.72.

Figure 13 shows time series of $IWC_{PSD}$ from the seven flights and estimated IWC from the two algorithms. As mentioned before, for each algorithm, a single set of fitting parameters is used for the combined data. Evidently, the method utilizing $Z_{dr}$ yields better results in term of estimation bias and rms (Fig. 13b and Table 1). In Fig. 14, estimation bias and std are expressed as a function of IWC. Note, that inclusion of $Z_{dr}$ improves estimation bias at all IWC points. On average, an improvement of

15 % in relative biases was achieved. As observed in Fig. 14, larger biases happen at IWC greater than 2 $gm^{-3}$. It is attributed to strong departures from the linear model in the joint IWC-$K_{dp}$ distribution. The inclusion of $Z_{dr}$ has been proved to be able to mitigate these large errors but not completely fix the problems. To improve the radar-derived IWC estimates further, more additional data processing (such as hydrometeorology classification) and/or more sophisticated regression models are needed.





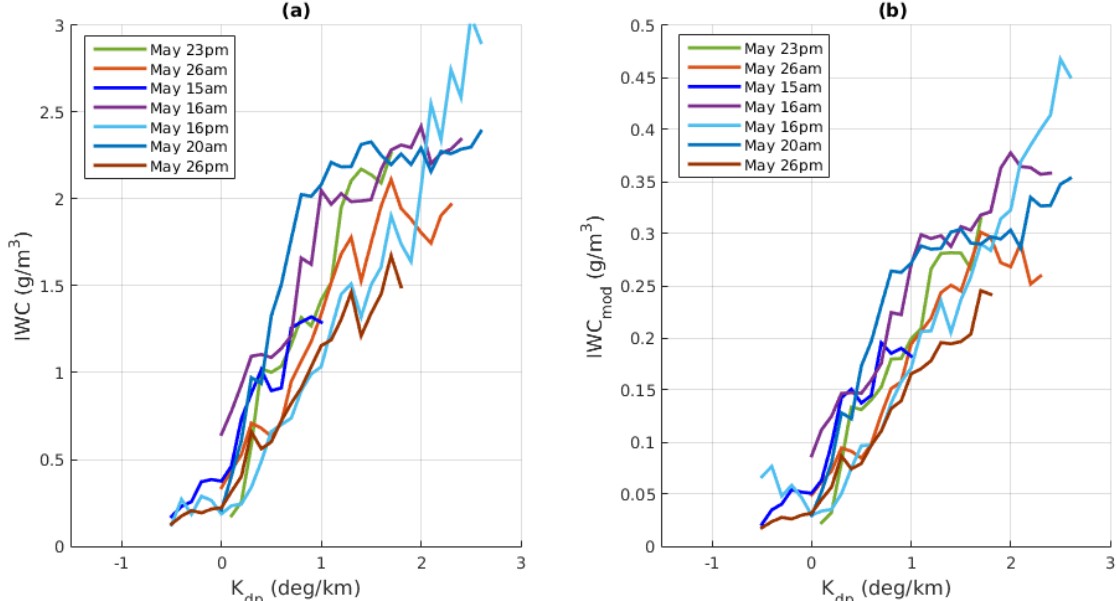

**Figure 12:** $IWC_{meas}$ (a) and $IWC_{mod}$ (b) as functions of $K_{dp}$ for the seven selected flights.

Table 1: Polarimetric methods performance for selected flights during the Cayenne 2015 campaign.

| Flight | $K_{dp}$ only | | | $K_{dp}$ and $Z_{dr}$ | | |
|---|---|---|---|---|---|---|
| | bias ($gm^{-3}$) | rms ($gm^{-3}$) | corr. coeff | bias ($gm^{-3}$) | rms ($gm^{-3}$) | corr. coeff |
| May 15am | -0.005 | 0.47 | 0.48 | 0.006 | 0.45 | 0.53 |
| May 16am | -0.088 | 0.46 | 0.85 | -0.016 | 0.42 | 0.87 |
| May 16pm | -0.048 | 0.37 | 0.77 | -0.024 | 0.30 | 0.83 |
| May 20am | 0.008 | 0.59 | 0.56 | 0.003 | 0.59 | 0.58 |
| May 23pm | -0.06 | 0.52 | 0.62 | -0.04 | 0.45 | 0.67 |
| May 26am | -0.002 | 0.48 | 0.64 | 0.006 | 0.45 | 0.70 |
| May 26pm | -0.025 | 0.46 | 0.62 | -0.024 | 0.42 | 0.70 |
| All* | -0.070 | 0.52 | 0.69 | -0.059 | 0.49 | 0.72 |

\* for all data points, optimal fitting parameters (0.903, 0.319) was used for $\boldsymbol{K_{dp}}$-only algorithm and (0.136, 0.037) was used

for ($\boldsymbol{K_{dp}}$, $\boldsymbol{Z_{dr}}$) algorithm.





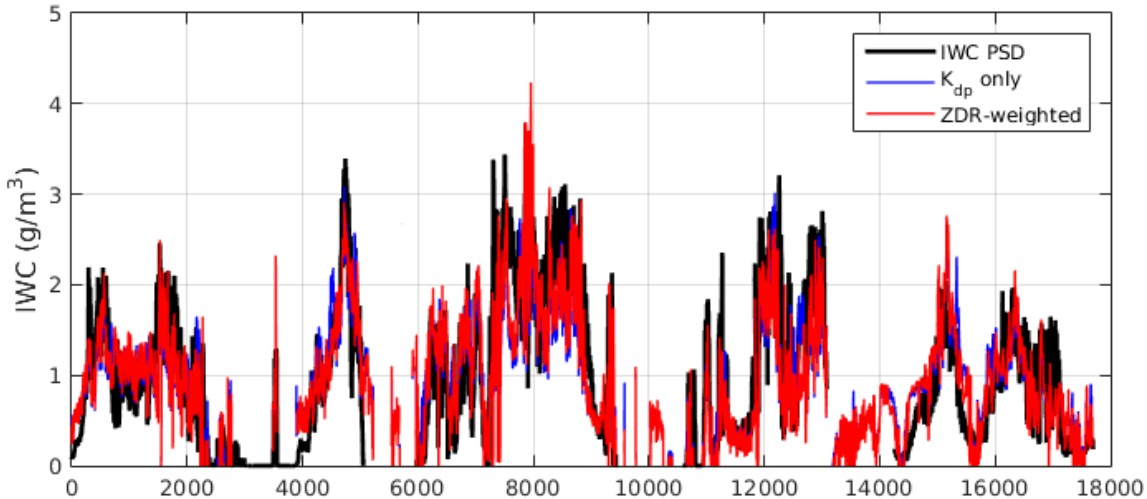

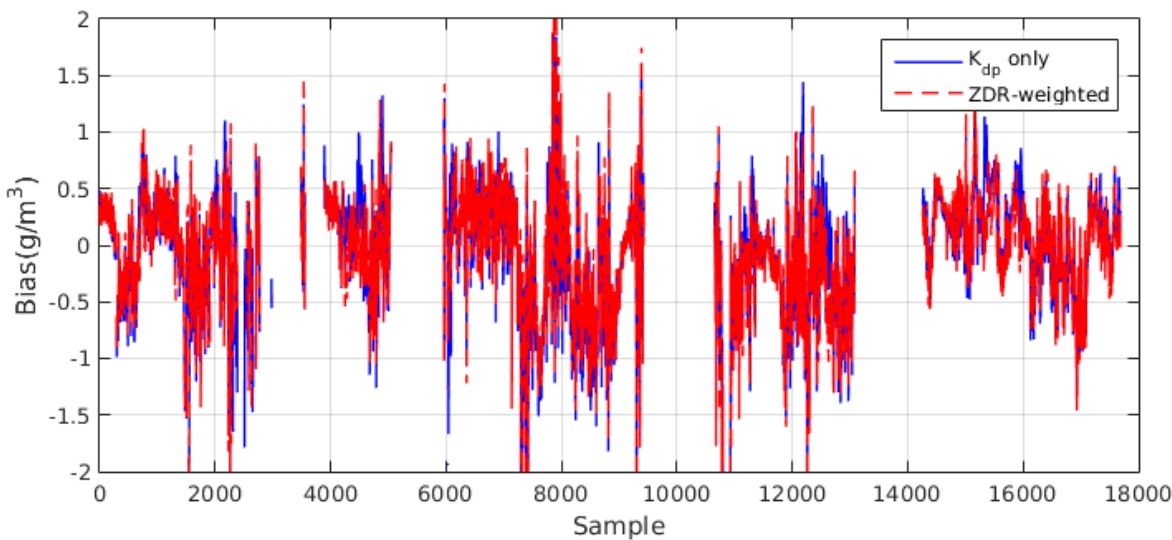

**Figure 13: (a) Combined IWC time series data from the selected flights: measured IWC (black line), estimated IWC using $K_{dp}$ alone (blue line) and estimated IWC using $K_{dp}$ and $Z_{DR}$ (red line). (b) Estimation errors for the two estimators.**



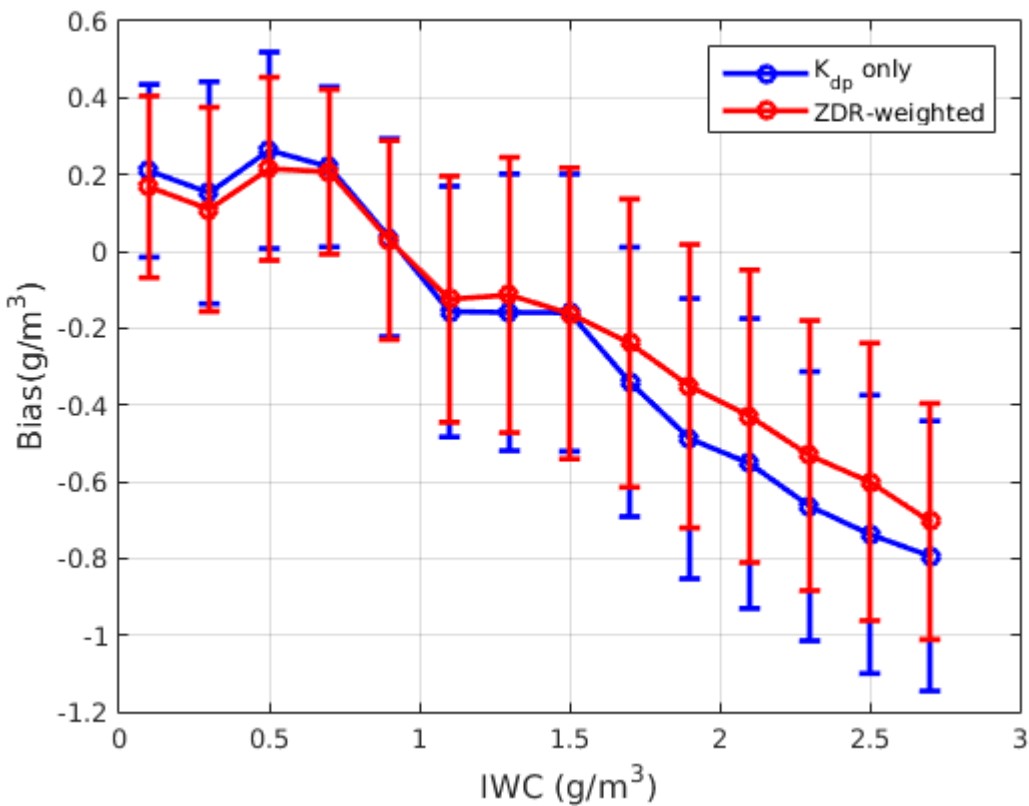

**Figure 14: Bias and rms difference as a function of IWC derived from the seven selected flights. Mean values and std are computed from data points in each $IWC$ bin of $0.2\ gm^{-3}$.**

## 7 Conclusion

Accurate detection and estimation of HIWC in tropical mesoscale convective systems are critical for reducing hazards caused by the ingestion of ice particles into the engines of commercial aircraft. The objective of this paper is to find a method to improve IWC retrieval from a side-looking X-band dual-polarization airborne radar. It is shown that the use of the specific differential phase ($K_{dp}$) and differential reflectivity ratio ($Z_{dr}$) significantly reduces errors in IWC retrieval over the conventional IWC-Z method. In general, IWC-$K_{dp}$ relationship can be approximated by a linear model and IWC retrieval

using $K_{dp}$ captures the IWC variation very well, regardless of the information of PSD. One major drawback of the $K_{dp}$ algorithm is that it provides large estimation biases when the ice particle's aspect ratio and/or orientation is changing. To mitigate this effect, $Z_{dr}$ is used to reduce the dependency of IWC on the variation of ice particles' shapes and orientation. We proposed a method, in which, IWCs are weighted by a function of $Z_{dr}$ before applying a linear model to the IWC-$K_{dp}$ joint distribution. This approach uses an assumption of constant particle mass within the radar volume. This is suitable for HIWC

regions which are often composed of very high density of small ice particles. $Z_{dr}$ at regions of mixtures of small pristine ice



crystals and larger particles such as aggregates is generally low ($\sim 0$ dB) and will not be used in the weighting function. Results from selected Convair-580 flights from the Cayenne campaign show that the proposed method is able to improve estimation biases by 15 % and rms difference by 6 %, on average. In our analysis, a single set of fitting parameters is applied for all the data points. The results can be improved further by including advanced data processing techniques such as ice crystal type

classification and/or using a more sophisticated regression model for the modified IWC-$K_{dp}$ joint distribution.

Most of HIWC data points used in this are measured at a narrow window of the temperature range ($-10\ °C \pm 2.5\ °C$). More data is needed to study the temperature variability of the proposed method.

*Acknowledgments.* This work is supported by the FAA (NAT-I-8417) and the NRC RAIR program. We would like to
acknowledge Jim Riley, Tom Bond and Chris Dumont of FAA and Steve Harrah of NASA for their support for this work. We thank the engineering, scientific and managerial staff from NRC and ECCC who made the project possible by working long hours during instrument integration and field operations. Special thanks to Ivan Heckman (ECCC) for support of processing cloud microphysical data. We would also like to acknowledge the support we received from the HAIC-HIWC community during the flight campaign.

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
