# Peer review of "Determination of Ice Water Content (IWC) in tropical convective clouds from X-band dual-polarization airborne radar"

_Atmospheric Measurement Techniques, 2019_

## Referee Comment (RC1) · Alexander Ryzhkov (Referee) · 14 Apr 2019

This is a very important study demonstrating a potential of a polarimetric radar for accurate estimation of ice water content in clouds by using a unique experimental setup combining X-band polarimetric radar measurements and in situ microphysical observations on the same airborne platform. It is shown in a number of flights that the polarimetric method grossly outperforms existing IWC-Z relations and that the combination of KDP and ZDR yields better accuracy of the IWC estimate than a use of sole KDP. I am particularly pleased to find out that empirically derived relations IWC(KDP) and IWC(KDP,ZDR) are very close to the theoretical relations derived by Ryzhkov et

al. (1998, 2018). I found that the multipliers a1 and a2 in the empirical formulas (12) and (13) (shown in Table 1) are within 6% of the theoretical ones which is remarkable. I think that this should probably be mentioned in the paper. There are several technical deficiencies in the manuscript which have to be addressed before the paper can be recommended for publication. (1) I was confused by the definitions of IWCmeas and IWCmod. It took me awhile to realize that IWCmod = (1-ZDR-1)IWCmeas. This is very weird and, I am sure, will confuse other readers as well. I would recommend simply using (1-ZDR-1)IWC instead of IWCmod in the text and labels in Figs. 5 and 12. (2) I may hypothesize that increasing negative bias in the radar IWC retrieval shown in Fig. 14 could be related to the minimal ZDR threshold of 0.6 dB. I would recommend to decrease the ZDR threshold in the IWC(KDP,ZDR) relation below 0.6 dB and see what happens. Adding large aggregates may disproportionally increase KDP and IWC. At the same time, ZDR decreases and may fall below 0.6 dB. Using values of ZDR lower than 0.6 dB will provide some "boost" for the IWC(KDP,ZDR) estimate. (3) Number concentration n in Eq (2) is not defined in the text. (4) In Eq (5), Kp2 is not equal to 0.177. It has to be the one for water. (5) Page 4. Cross sections $\sigma$hh,vv are not used in Eqs (1) – (6). (6) Page 5. The approximation (1-ZDR-1) IWC $\approx$ KDP is not correct and is not consistent with the value a2 = 0.135 shown in Table 1. (7) The reference to Korolev et al. (2018) can not be found in the reference list. (8) Page 8. Both $\Phi$DP and $\Psi$DP may exhibit discontinuities due to phase wrapping. (9) Page 9. Two equations IWC(Z) are very different and both differ much from the popular Hogan et al. IWC(Z) equation. Please clarify and comment. (10) English usage has to be improved, e.g., data are plural, not single, etc. References Ryzhkov, A., P. Bukovcic, A. Murphy, P. Zhang, and G. McFarquhar, 2018: Ice microphysical retrievals using polarimetric radar data. 10th European Conference on Radar in Meteorology and Hydrology, 1 – 6 July, The Netherlands, # 40. Available online at: projects.knmi.nl/erad2018/ERAD2018_extended_abstract_040.pdf.

---

## Referee Comment (RC2) · Anonymous Referee #2 · 2 May 2019

This manuscript presents an assessment of ice water content (IWC) retrieval algorithms based on KDP only and combined KDP-ZDR observations using measured IWC from aircraft in-situ probes. Results show that both the KDP-only and combined KDP-ZDR method works much better than the reflectivity-based retrieval algorithms. Although KDP-only and KDP-ZDR combined IWC retrieval algorithms are not new concept, an algorithm derived from quasi-collocated in-situ measurement and radar observations is a good addition to the existing literature. Some sections of the manuscript lack details and are somewhat difficult to follow, especially section 2. Modifications to the manuscript are needed before it can be recommended for publication. Major concerns and questions that needed to be addressed are outlined below.

General Comments:

1. Combining KDP and ZDR in IWC retrieval is not new concept. For example, Eq. (13) in the current manuscript is very similar to Eq. (29) in Ryzhkov et al. (1998). In my opinion, the authors should discuss more on the relationships and differences between the current study and the other more theoretical studies.

2. Section 2 lacks detailed derivation and/or reference to existing literatures and is very difficult to follow. This section needs to be heavily rewritten. The author should provide detailed derivations or clearly refer to existing literatures for the equations. When doing so, the authors should make sure that the conventions of the equations are consistent.

3. The parameters (a, b) used in the Kdp only and Kdp-ZDR combined algorithm are obtained from linear fittings of data from 7 cases. By looking at Figure 12, it seems different values of (a, b) could be obtained if data from only one, or some of, the 7 cases are used in the fitting. What is the variability of the parameters (a, b)? How large the retrieval uncertainty will be due to the uncertainty in (a, b)?

Specific comments: P2, L14: What is the conclusion of Ryzhkov et al. (1998)?

P3, L11 and L14: Which convention is used for the back-scattering matrices and forward scattering amplitudes?

P3, L10 and L13: The radar observables, Zdr and Kdp, are usually integrated over a particle size distribution. Although Eq. (1) and Eq. (2) are OK if all the particles are of the same size and aspect ratio, this situation rarely happens. I think these equations can be removed since Eq. (5) and Eq. (6) are more general. Instead, the authors can discuss scattering properties in single particle level here.

P3, L15-17: This sentence is confusing. I guess the authors mean that Zdr does not change with increasing number of small particles while Kdp increases with increasing number of particles. The authors may want to rephrase this sentence.

P3, L20: Is the permittivity of particle based on solid ice?

P5, Eq. (9-11): By using symbol for "approximately equal", do the authors really mean "proportional"?

P5, Eq. (11): Derivation for this equation is needed. Also, Eq. (9) and Eq. (10) use integration over a PSD, why Eq. (11) only use one mass?

P5, L7-9: Again, detailed derivation is needed. How to derive a closed form of IWC as a function of Kdp and ZDR? What does "particle mass variation is small within the radar volume" mean?

P5, L19-20: What are the physical meanings of the constants b1 and b2?

P10, L7: What does "initial observations" mean? Should the "include" be replaced with "indicate"?

P10, L8: "latter" is confusing. Better to clearly state which observations are weighted and biased towards Zdr.

P10, L20: No blue line in Fig. 5(c).

P10, L7-12: Are the polarimetric variables Kdp, Zdr, Zh, and rho_hv shown in Fiugre (4) measured at a specific distance from the airplane or averaged over a range? How about those shown in Figure (5) and (6)?

P13, L6: What are the coefficients (a, b) used for Kdp-only and (Kdp, ZDR) algorithms? Are same values used for the other case?

P20, L12: I think the word "significantly" is too subjective and optimistic. For example, when Kdp=1, in panel (a), IWC varies by 50% of the maximum value (range 1~2 with maximum 2), and in panel (b), modified IWC also varies by 50% (range 0.15~0.3 with maximum 0.3). I would suggest remove the word "significantly".

Technical corrections: P1, L9: Is the differential reflectivity here in linear scale or log scale?

P2, L10: Aydin and Tang (1995) should be (1997).

P3, L8: The book of Bringi and Chandrasekar (2001) has over 600 pages. Detailed page numbers or equation numbers are needed. The same for P3 L22 and other places.

P4, L26: IWC is not sensitive to shape and orientation sounds strange. IWC is independent of them. The authors may want to rephrase this sentence.

P11, Figure 3 caption: "Bottom panel is shows" remove "is".

P13, Figure 5 caption: using two values (0.1 and 0.05 degree) and "respectively" for three panels is confusing.

P12, L5: "blue" should be "black" for measured IWC.

P14, L5: "estimations" should be "estimation".

P14, L5: The panels of Figure 6 are not labeled by 'a' or 'b'. Same for Figure 11.

P15, Figure 6: Should vertical axis labeled as "Error" instead of "Bias"?

P18, L8: No ellipses in Fig. 11a.
* * *

---

## Referee Comment (RC3) · Anonymous Referee #3 · 8 May 2019

General Comment

The manuscript compared three radar measurement based IWC retrieval methods (Z only based, Kdp only based, and Kdp and Zdr based) using X-band polarimetric observation and in-situ observation and demonstrated that the Kdp and Zdr based method can provide better IWC estimates. The retrieved IWC agreed very well with the observed IWC from the in-situ measurement. The methods are based on observations, and I would expect to extend the techniques to other places in future; however, the manuscript did not mention effectiveness of the methods for other cloud types at different locations. The subject and results of the manuscript are suitable for the scope of

the journal. However, this manuscript lacked detailed descriptions about how to estimate coefficients. In particular, estimated coefficient values were not presented. Some figure captions lacked explanations. I recommend major revision in terms of comments below.

Specific comments

1. The manuscript lacked detailed descriptions about how to estimate coefficients ($a_1$, $b_1$, $a_2$, and $b_2$) and their values. Information below is at least needed. 1) The estimated coefficient values. 2) Were the coefficient values estimated for each case or constant for all cases? 3) How did you sample radar data? In-situ data was valid near the aircraft, while the radar could not collect data below a range of 1000 m from the aircraft. What are locations of the radar data used for the coefficient estimations? How did you match the radar data and the in-situ data? The sampling method may control the accuracy of the linear regression technique. 4) How many cases (how many hours) were used for coefficient estimates? 5) Is there a dependency of the estimated coefficients on cases? 6) It is very worth to add a flow chart of the IWC retrieval technique.

2. Details about the radar data are needed. What are the beamwidth, radar range gate spacing, and time resolution? Did the radar point at a fixed direction during the flights or scan some directions (in that case, what are the elevation and azimuth)? Did the radar sampling volume match the in-situ measurement sampling volume?

3. A more description about IWC from PSD is also needed. What D-M parameterization was used? How was it tuned using IKP2 measurements? Again, a flow chart of the IWC calculation should be useful. Did the image probes scan particles at a horizontal plane (parallel to the horizontal plane) or vertical plane (perpendicular to the horizontal plane)? Can the scan direction cause some errors in the IWC calculation?

4. Is the technique presented in the manuscript valid for cases where graupel existed? Graupel particles generally have aspect ratio $\sim 1$, which are not sensitive to Kdp or Zdr. Deep convective cases could include supercooled liquid droplets and heavilyrimed particles. Were there effects of supercooled liquid droplets and heavily-rimed particles in the cases used in the analysis? Was the case dependency of the proposed technique? More case descriptions would be needed to help understanding the case dependency.

5. P. 14, lines 2-3: This sentence does not make sense to me. Please give more description about why Kdp can improve large aggregates effect, although it is sensitive small crystals. Just before the sentence, it was stated that "ðÌŘ¿ðÍŚŚðÍŚÍ is more sensitive to the oriented small ice crystals." Why Kdp can improve large aggregates effect? How did you know the orientation of ice particles?

6. P. 18 lines 10-11 "On the other hand,...": This sentence does not make sense to me. Why the modified IWC was less sensitive to the particles' shape and orientation, even though the Zdr constraint well worked for the IWC estimate?

7. Evaluations for mode error sources should be discussed; how could Kdp estimation resolution, Zdr bias, and in-situ instrument limitation (difference between IWCmeas and IWCpsd) affect the IWC retrievals?

8. Fig. 12: I do not know the meaning of Fig. 12. Why can you see "the modified version of IWC is better replicated by a simple linear regression model" from this figure? "Modified IWC" is the just measured IWC scaled by Zdr. Why is this better replicated by a simple linear regression model then measured IWC? In both Fig. 12a and Fig. 12b, it seems that there is ∼50% variability in IWC at Kdp = 1. There is no significant difference between Fig. 12a and Fig. 12b. If you wanted to say that Eq. 13 can provide better correlations between IWC and polarimetric variables (Kdp and Zdr), this has been already shown by R^2 values in Fig. 5.

9. "Modified IWC" is a confused phrase. In the manuscript, "modified IWC" represents the left hand side of Eq. 13. (1-ZDR^-1)*IWC. This is IWC scaled by [1-ZDR^-1], not actual IWC. Use an appropriate phrase through the text.

Minor comments

1. Did you do attenuation corrections for reflectivity and Zdr? Attenuation corrections may not be needed for pure ice precipitation, but it would be good to mention about this.

2. Did you do calibrate Zdr values for systematic offset?

3. What was the window size for calculating Kdp? What is the special resolution of Kdp?

4. P. 10, lines 1-2 "The radar estimates...": This sentence is unclear. Please add detailed descriptions.

5. P. 10, line 8 (2): This feature is unclear in Fig. 4. Scatter plots of MMD, RHOhv and Zdr should be helpful.

6. P. 10, line 9 and p. 14, line 2: How did you know the orientation of small crystals? Kdp should also be sensitive to larger crystals with aspect ratio /= 1. How did you know the particle aspect ratios?

7. P. 10, lines 10-11: In Fig. 5, it seems to me that the break point where Zdr started increasing and RHOhv deceased was at Kdp~1.5 deg/km.

8. P. 10, line 20: I cannot see any blue line Fig. 5.

9. P. 10, lines 20-21: I cannot see measured IWC or modified IWC in Fig. 5.

10. Fig. 3b and Fig. 7b: Why is there a break line at 6-7 km range in reflectivity fields?

11. Please give height information for Figs. 3, 7, 9, 10.

12. Please give information about height, range, and temperature for Figs. 4, 5, 6, 8, 11, 13.

13. Unit of Kdp should be degrees / km through the manuscript?

14. Fig. 5: What do red lines represent?

15. Fig. 5: I confused 'scatter plots' with red and black dots. I guess that 'scatter plots' here meant color shades representing frequency distributions. I recommend using 'frequency distribution' instead of 'scatter plot.'

16. Fig 5: missed (a), (b), and (c)

17. P. 18 line 8: I cannot find ellipses in Fig. 11.

18. Caption of Fig. 3: Delete "is"

---

## Author Comment (AC1) · 11 Jul 2019

**Response to the Referee 1 comments**

This is a very important study demonstrating a potential of a polarimetric radar for accurate estimation of ice water content in clouds by using a unique experimental setup combining X-band polarimetric radar measurements and in situ microphysical observations on the same airborne platform. It is shown in a number of flights that the polarimetric method grossly outperforms existing IWC-Z relations and that the combination of KDP and ZDR yields better accuracy of the IWC estimate than a use of sole KDP. I am particularly pleased to find out that empirically derived relations IWC(KDP) and IWC(KDP,ZDR) are very close to the theoretical relations derived by Ryzhkov et al. (1998, 2018). I found that the multipliers a1 and a2 in the empirical formulas (12) and (13) (shown in Table 1) are within 6% of the theoretical ones which is remarkable. I think that this should probably be mentioned in the paper. There are several technical deficiencies in the manuscript which have to be addressed before the paper can be recommended for publication.

We would like to thank the referee for these very helpful and constructive comments. We have highlighted the significance of our experimental result as you suggested. We have also addressed the various deficiencies that are pointed out by the referee throughout the paper. Please find our detailed responses to your comments as follows.

(1) I was confused by the definitions of IWCmeas and IWCmod. It took me awhile to realize that IWCmod = (1-ZDR-1)IWCmeas. This is very weird and, I am sure, will confuse other readers as well. I would recommend simply using (1-ZDR-1)IWC instead of IWCmod in the text and labels in Figs. 5 and 12.

We agree with the referee's comment. In the revised manuscript, IWCmod is replaced by $(1_{Z_{DR}}^{-1})IWC$ to avoid any confusion.

(2) I may hypothesize that increasing negative bias in the radar IWC retrieval shown in Fig. 14 could be related to the minimal ZDR threshold of 0.6 dB. I would recommend to decrease the ZDR threshold in the IWC(KDP,ZDR) relation below 0.6 dB and see what happens. Adding large aggregates may disproportionally increase KDP and IWC. At the same time, ZDR decreases and may fall below 0.6 dB. Using values of ZDR lower than 0.6 dB will provide some "boost" for the IWC(KDP,ZDR) estimate.

We thank the referee for this very good suggestion. We have tested and found out that reducing the $Z_{DR}$ threshold indeed improve the estimation bias. However, smaller $Z_{DR}$ threshold increase rms as $(1 - Z_{DR}^{-1})$ comes close to zero. Thus, we selected the threshold at which rms of Kdp-only method and $(K_{dp}, Z_{DR})$ method are equal. We have revised the text and added figures to include this change.

(3) Number concentration n in Eq (2) is not defined in the text.

The definition and unit of variable n in Eq. (2) is added.

(4) In Eq (5), Kp2 is not equal to 0.177. It has to be the one for water.

We thank the referee for pointing out this error. The dielectric factor of water at $0^o$ C is used for the computation of equivalent reflectivity, $K_p^2 = 0.93$

(5) Page 4. Cross sections σhh,vv are not used in Eqs (1) – (6).

We agree. The radar cross sections are now removed.

(6) Page 5. The approximation (1-ZDR$^{-1}$)IWC ≈KDP is not correct and is not consistent with the value a2 = 0.135 shown in Table 1.

We thank the referee for pointing this out. We actually meant "proportional" instead of "approximately equal".

(7) The reference to Korolev et al. (2018) cannot be found in the reference list.

The reference to Korolev et al. (2018) has been added.

(8) Page 8. Both ΦDP and ΨDP may exhibit discontinuities due to phase wrapping.

We agree. Correction has been made.

(9) Page 9. Two equations IWC(Z) are very different and both differ much from the popular Hogan et al. IWC(Z) equation. Please clarify and comment.

The two equations were derived by fitting a simple power-law curve to the joint frequency distribution of IWC and reflectivity (similar to Eq. (1) in Protat *et al.* (2016) but for measured X-band reflectivity in the HIWC flights). There might be a small bias in the radar reflectivity calibration so the coefficients in those equations could present a small error. However, the IWC(Z) is included just to demonstrate the traditional approach that uses Z can lead to large uncertainties in the HIWC regions.

(10) English usage has to be improved, e.g., data are plural, not single, etc.

We thank the referee for this comment. We have tried to correct language errors and improved the manuscript.

References:

Ryzhkov, A., P. Bukovcic, A. Murphy, P. Zhang, and G. McFarquhar, 2018: Ice microphysical retrievals using polarimetric radar data. 10th European Conference on Radar in Meteorology and Hydrology, 1 – 6 July, The Netherlands, # 40. Available online at: projects.knmi.nl/erad2018/ERAD2018_extended_abstract_040.pdf.

Thanks for this reference. It has been added to the revision.

---

## Author Comment (AC2) · 11 Jul 2019

**Response to the Referee 2 comments**

This manuscript presents an assessment of ice water content (IWC) retrieval algorithms based on KDP only and combined KDP-ZDR observations using measured IWC from aircraft in-situ probes. Results show that both the KDP-only and combined KDP-ZDR method works much better than the reflectivity-based retrieval algorithms. Although KDP-only and KDP-ZDR combined IWC retrieval algorithms are not new concept, an algorithm derived from quasi-collocated in-situ measurement and radar observations is a good addition to the existing literature. Some sections of the manuscript lack details and are somewhat difficult to follow, especially section 2. Modifications to the manuscript are needed before it can be recommended for publication. Major concerns and questions that needed to be addressed are outlined below.

We would like to thank the reviewer for many constructive comments which helped improve the manuscript greatly. We have made many changes in the revised manuscript, and have added more references to improve the presentation of the manuscript. Please find our responses to your comments as follows.

General Comments:

1. Combining KDP and ZDR in IWC retrieval is not new concept. For example, Eq. (13) in the current manuscript is very similar to Eq. (29) in Ryzhkov et al. (1998). In my opinion, the authors should discuss more on the relationships and differences between the current study and the other more theoretical studies.

We agree that combining KDP and ZDR in IWC retrieval is not a new concept. As a result, we didn't really provide an in depth review. In literature, there are many different equations representing the relationships between IWC, Z, $K_{dp}$, $Z_{dr}$ and parameters of the particle size distribution. We tested most of them and only focused on the methods (Kdp-only and $(K_{dp}, Z_{DR})$ combination) that work best for our data sets in the HIWC environment. We believe we have included enough review of the concept and we have also added a recent reference (Ryzhkov et al. (2018)) in the revised version of the manuscript.

2. Section 2 lacks detailed derivation and/or reference to existing literatures and is very difficult to follow. This section needs to be heavily rewritten. The author should provide detailed derivations or clearly refer to existing literatures for the equations. When doing so, the authors should make sure that the conventions of the equations are consistent.

In the revised manuscript, we have added references for each equation in section 2. Part of section 2.2 (P5) was re-written. Again, because the concept is not new, we did not include all the derivations that can be found in the literatures. Instead, we summarized the main findings which are important to our study.

We thank the referee for the comment on the inconsistence of the convention of the equations. We have corrected the errors.

3. The parameters (a, b) used in the Kdp only and Kdp-ZDR combined algorithm are obtained from linear fittings of data from 7 cases. By looking at Figure 12, it seems different values of (a, b) could be obtained if data from only one, or some of, the 7 cases are used in the fitting. What is the variability of the parameters (a, b)? How large the retrieval uncertainty will be due to the uncertainty in (a, b)?

In the revised manuscript, we have included the fitting coefficients for both methods in Table 2. The standard deviations for ($a_1$, $b_1$) for the $K_{dp}$-only method and for ($a_2$, $b_2$) for the ($K_{dp}, Z_{DR}$) method are (0.12, 0.33) and (0.032, 0.033), respectively. The uncertainty of the retrieval depends on the uncertainty in (a, b) and the values of $K_{dp}$ and $Z_{DR}$. Typical values of $K_{dp}$ and $Z_{DR}$ for HIWC regions (MMD between 0.25 mm to 0.8 mm) are about 1 deg/km and 1.12 (or 0.5 dB) (from Fig. 6 in the revision). At those typical values, standard deviation of IWC estimates using ($K_{dp}, Z_{DR}$) is 0.6 g/m$^3$.

Specific comments:

P2, L14: What is the conclusion of Ryzhkov et al. (1998)?

The conclusion of Ryzhkov et al. (1998) is briefly summarized in Section 2. In the revision, the text has been modified and a new reference (Ryzhkov et al. (2018)) has been added.

P3, L11 and L14: Which convention is used for the back-scattering matrices and forward scattering amplitudes?

It is in forward scatter alignment (FSA) convention. We thanks the referee for pointing this out. Additional text has been added to the revision to make it clear.

P3, L10 and L13: The radar observables, Zdr and Kdp, are usually integrated over a particle size distribution. Although Eq. (1) and Eq. (2) are OK if all the particles are of the same size and aspect ratio, this situation rarely happens. I think these equations can be removed since Eq. (5) and Eq. (6) are more general. Instead, the authors can discuss scattering properties in single particle level here.

We thanks the referee for this comment. The Eq. (1) and (2) are for n particles of same size D and axis ratio $r$. The purpose of including Eq. (1) and (2) is to show that large number ice particles (which is relevant to HIWC regions) can contribute to a significant $K_{dp}$. We'd like to keep those equations instead of including a discussion on the particle's scattering properties because this paper mainly focuses on an empirically derived estimator for IWC based on radar polarimetric parameters ($K_{dp}, Z_{DR}$).

P3, L15-17: This sentence is confusing. I guess the authors mean that Zdr does not change with increasing number of small particles while Kdp increases with increasing number of particles. The authors may want to rephrase this sentence.

We agree with the referee. The sentence has been rephrased.

P3, L20: Is the permittivity of particle based on solid ice?

It is the relative permittivity of ice particles in the radar volume, which is    unknown. An approximate relationship between relative permittivity and density of snow/ice particle can be found in Matrosov *et al.* (1996).

P5, Eq. (9-11): By using symbol for "approximately equal", do the authors really mean "proportional"?

We thank the referee for pointing this out. The error has been corrected.

P5, Eq. (11): Derivation for this equation is needed. Also, Eq. (9) and Eq. (10) use integration over a PSD, why Eq. (11) only use one mass?

(please see the response after the next question)

P5, L7-9: Again, detailed derivation is needed. How to derive a closed form of IWC as a function of Kdp and ZDR? What does "particle mass variation is small within the radar volume" mean?

In the revised manuscript, we have removed Eqs (9)-(11) and revised the text in L7-9. We've also added a recent reference by Ryzhkov et al. (2018). Detailed theoretical derivation and discussion of the IWC approximations based on $(K_{dp}, Z_{DR})$ can be found in Ryzhkov et al. (1998, 2018); therefore, we decided not to include them in the paper.

P5, L19-20: What are the physical meanings of the constants b1 and b2?

We thank the reviewer for this question. Mathematically, the intercepts (b1 and b2) are mean values of IWC at $K_{dp} = 0°$ . Physically, we do not know their meaning. When the ice particle orientation uniformly distributed in a plane perpendicular to the radar beam, $K_{dp} = 0°$ but IWC can be larger than zero.

P10, L7: What does "initial observations" mean? Should the "include" be replaced with "indicate"?

We have modified the text to remove the confusion. It now reads "From Fig. 4, it follows: …"

P10, L8: "latter" is confusing. Better to clearly state which observations are weighted and biased towards Zdr.

The sentence has been modified as suggested.

P10, L20: No blue line in Fig. 5(c).

We thank the referee for pointing this error. It is now corrected.

P10, L7-12: Are the polarimetric variables Kdp, Zdr, Zh, and rho_hv shown in Fiugre (4) measured at a specific distance from the airplane or averaged over a range? How about those shown in Figure (5) and (6)?

Those radar parameters were measured at a distance of 1000 m from the aircraft. The data were not averaged over the range but were decimated in time (along the flight path) to match with the temporal resolution of the in-situ data. Additional information on this has been included in the revised manuscript.

P13, L6: What are the coefficients (a, b) used for Kdp-only and (Kdp, ZDR) algorithms? Are same values used for the other case?

In study case May 26, the coefficients (a, b) for Kdp-only and (Kdp, ZDR) algorithms are (0.94, 0.7) and (0.12, 0.07), accordingly. The coefficients are different and are optimized for each case. We have added estimated coefficients for all selected flights in Table 2 and discussed on the variability of those coefficients in section 6.

P20, L12: I think the word "significantly" is too subjective and optimistic. For example, when Kdp=1, in panel (a), IWC varies by 50% of the maximum value (range 1~2 with maximum 2), and in panel (b), modified IWC also varies by 50% (range 0.15~0.3 with maximum 0.3). I would suggest remove the word "significantly".

We completely agree with the referee. The word "significantly" has been removed.

Technical corrections:

P1, L9: Is the differential reflectivity here in linear scale or log scale?

It is in log scale.

P2, L10: Aydin and Tang (1995) should be (1997).

We apologize for the error. It has been corrected.

P3, L8: The book of Bringi and Chandrasekar (2001) has over 600 pages. Detailed page numbers or equation numbers are needed. The same for P3 L22 and other places.

We have added equation numbers in the book of Bringi and Chandrasekar (2001) to the revised manuscript.

P4, L26: IWC is not sensitive to shape and orientation sounds strange. IWC is independent of them. The authors may want to rephrase this sentence.

We thank the referee for this comment. This sentence has been revised. It now reads "For a given radar volume, if the orientation of the ice crystal changes, $K_{dp}$ value changes (Eq. (7)) while the IWC of the radar volume is not."

P11, Figure 3 caption: "Bottom panel is shows" remove "is".

This typo has been corrected.

P13, Figure 5 caption: using two values (0.1 and 0.05 degree) and "respectively" for three panels is confusing.

The values are for the last two panels (panel b and c). The caption has been modified to avoid the confusion.

P13, L5: "blue" should be "black" for measured IWC.

The error has been corrected.

P14, L5: "estimations" should be "estimation".

We thank the referee for pointing this error. It has been corrected.

P14, L5: The panels of Figure 6 are not labeled by 'a' or 'b'. Same for Figure 11.

We apologize for the missing labels. In the revised manuscript, all the figures have been checked and missing labels have been added.

P15, Figure 6: Should vertical axis labeled as "Error" instead of "Bias"?

In this study, measured IWC is considered as ground truth, hence, the difference between estimated IWC from radar data and measured IWC is labeled as bias. However, we have corrected the second sentence in the caption of Fig. 6 in the revision: "Bottom panel shows estimation () biases for the three estimators."

P18, L8: No ellipses in Fig. 11a.

We thank the reviewer for pointing this out. Ellipses have been added to Fig. 11a (it is now Fig. 12 in the revision).

---

## Author Comment (AC3) · 11 Jul 2019

**Response to the Referee 3 comments**

**RC#3:**

General Comment

The manuscript compared three radar measurement based IWC retrieval methods (Z only based, Kdp only based, and Kdp and Zdr based) using X-band polarimetric observation and in-situ observation and demonstrated that the Kdp and Zdr based method can provide better IWC estimates. The retrieved IWC agreed very well with the observed IWC from the in-situ measurement. The methods are based on observations, and I would expect to extend the techniques to other places in future; however, the manuscript did not mention effectiveness of the methods for other cloud types at different locations. The subject and results of the manuscript are suitable for the scope of the journal. However, this manuscript lacked detailed descriptions about how to estimate coefficients. In particular, estimated coefficient values were not presented. Some figure captions lacked explanations. I recommend major revision in terms of comments below.

We thank the referee for his/her review and encouraging comments. Regarding the suitability of our method to other non-HIWC environments, we focused our analysis to HIWC environment so didn't speculate on if our result is applicable to other conditions. We are planning to extend our analysis of large dataset we accumulated in a recent projects and will report our findings once we complete the work.

As outlined in our responses the two other reviewers, we have made significant changes in the manuscript and added new figures and tables. We hope these changes make the paper easier to follow. Please find our responses to your specific comments as follows.

Specific comments

1. The manuscript lacked detailed descriptions about how to estimate coefficients (a1, b1, a2, and b2) and their values. Information below is at least needed:

1) The estimated coefficient values

2) Were the coefficient values estimated for each case or constant for all cases?

We thank the referee for pointing this out. In the revision, we have added texts to describe how to estimate the fitting parameters (in the caption of Fig. 5). Values of those parameters were estimated for each study cases are given in Table 2 of the revision. They were also computed for combined data of seven selected flights. The variability of those fitting parameters is briefly discussed in section 6.

3) How did you sample radar data? In-situ data was valid near the aircraft, while the radar could not collect data below a range of 1000 m from the aircraft. What are locations of the radar data used for the coefficient estimations? How did you match the radar data and the in-situ data? The sampling method may control the accuracy of the linear regression technique.

As provided in section 3.1, radar profiles were extracted at a horizontal distance of 1000 m from the aircraft. The data rate of the processed in-situ microphysics data is much lower than the radar data. Hence, the radar data were decimated to match with temporal resolution of the in-situ data (e.g. 1 s). At the Convair-580 average ground speed of 100 $ms^{-1}$, this results in a 100 m resolution along the flight line. The radar range resolution is 75 m and the data were not average over range.

4) How many cases (how many hours) were used for coefficient estimates?

It was 4.91 hours of data collection time (17699 data point at 1 s sampling period) for the seven selected cases.

5) Is there a dependency of the estimated coefficients on cases?

We do not see a dependency of the estimated coefficients over the study cases.

6) It is very worth to add a flow chart of the IWC retrieval technique.

We thank the reviewer for this good suggestion. However, we haven't added additional flowchart as we felt the processes of our analysis has been detailed in the revised manuscript. Also, it may be more adequate to have the flowchart of the IWC retrieval technique when we complete a work where additional steps are added to the technique to improve its performance further.

2. Details about the radar data are needed. What are the beamwidth, radar range gate spacing, and time resolution? Did the radar point at a fixed direction during the flights or scan some directions (in that case, what are the elevation and azimuth)? Did the radar sampling volume match the in-situ measurement sampling volume?

We thank the referee for this comment. We have added a table (Table 1 in the revised manuscript), which lists some important radar parameters. All the antennas of the X-band radar system are fixed. As mentioned in section 3, the radar data used in this study were sampled at 1000 m horizontal distance from the aircraft, whereas the in-situ data were measured right at the aircraft location. For in-situ probes, the sampling volume is much smaller compared to the radar sampling volume.

3. A more description about IWC from PSD is also needed. What D-M parameterization was used? How was it tuned using IKP2 measurements? Again, a flow chart of the IWC calculation should be useful. Did the image probes scan particles at a horizontal plane (parallel to the horizontal plane) or vertical plane (perpendicular to the horizontal plane)? Can the scan direction cause some errors in the IWC calculation?

IWC was calculated from composite 2DS+PIP PSDs based on $M = aD^b$ size-to-mass parameterization. We agree that a flowchart of IWC calculation should be useful but it is beyond the scope of this work and should be found in the IWC reference paper (Korolev *et al.* 2018). The coefficient a and b were tuned using a non-linear regression to find the best correlation between IWC measured by IKP2 and calculated from PSD. The coefficients a and b are determined by dominant particle habits, and therefore they may vary depending on type of clouds, temperature range and altitude were the particles were sampled. For the particle images sampled from the Convair580 during the HIWC campaign the best agreement between two ways of IWC measurements were obtained for a=7.0044e-12 and b=2.3. The laser beams of both PIP and 2DS probes used for particle image sampling were oriented vertically (the horizontal 2DS channel was not used). Such arrangement provides measurements of the maximum projection of particle images. The direction of scanning may affect assessment of IWC from 2D particle imagery. However, tuning of the a and b coefficients, which was employed in this study, compensates the effect of the direction of particle image sampling.

4. Is the technique presented in the manuscript valid for cases where graupel existed? Graupel particles generally have aspect ratio ~1, which are not sensitive to Kdp or Zdr. Deep convective cases could include supercooled liquid droplets and heavily- rimed particles. Were there effects of supercooled liquid droplets and heavily-rimed particles in the cases used in the analysis? Was the case dependency of the proposed technique? More case descriptions would be needed to help understanding the case dependency.

As briefly mentioned in the manuscript, the fraction of flight segments with supercooled drops was <5%.  Our data are also consistent with a multi-year and multi-aircraft observations of the HAIC-HIWC flight data that are detailed in FAA report (Strapp et. al, 2018).  We didn't try to filter out those data points with supercooled drops in our analysis.  Having said that we agree with the reviewer that the convective clouds we flew are likely consisted of heavily rimed particles and graupel at the convective cores of the storms we flew.

Our main goal is to investigate if adding polarimetric capability to operational pilot X-band radars allows to detect and avoid HIWC environment.  In this regard, commuter aircraft typically avoid the convective cores based of high reflectivity values and other data. So our work and conclusion don't extend to convective core areas.

5. P. 14, lines 2-3: This sentence does not make sense to me. Please give more description about why Kdp can improve large aggregates effect, although it is sensitive small crystals. Just before the sentence, it was stated that (???) is more sensitive to the oriented small ice crystals." Why Kdp can improve large aggregates effect? How did you know the orientation of ice particles?

We agree with the referee on this comment. We've added information about $K_{dp}$ and $Z_{dr}$ as a function of MMD (Fig. 6) in the revised manuscript. The text also has been modified in the revised manuscript: "The large errors in the $IWC - Z$ estimator are due to the presence of mixtures of large aggregates and small ice crystal regions as indicated in the PIP images (not shown) in clouds.  Large aggregates have a dominant contribution into the radar reflectivity, which explains the positive biases of the $IWC - Z$ estimates. On the other hand, the magnitude of $K_{dp}$ in aggregates with MMD > 2 mm is usually smaller than 0.4 °/km and in small ice particles (MMD in the range 0.3 – 1 mm) is between 0.6 °/km  to 1 °/km (Fig. 6a).  It follows that estimators utilizing  $K_{dp}$ information may not have strong biases toward large aggregates in radar volumes like the $IWC - Z$ estimator."

6. P. 18 lines 10-11 "On the other hand,. . .": This sentence does not make sense to me. Why the modified IWC was less sensitive to the particles' shape and orientation, even though the Zdr constraint well worked for the IWC estimate?

We thank the reviewer for this comment. The sentence has been modified in the revised manuscript to have a better explanation: "On the other hand, the product $(1 - Z_{DR}^{-1})IWC$ already includes the particles' shape and orientation effects, thus, estimates based on it yield should better results."

7. Evaluations for mode error sources should be discussed; how could Kdp estimation resolution, Zdr bias, and in-situ instrument limitation (difference between IWCmeas and IWCpsd) affect the IWC retrievals?

We agree with the referee that the error analysis of the retrieval algorithms is important. In addition to the factors mentioned by the referee, errors in IWC retrievals also depends on the uncertainty of the fitting coefficients. A complete error analysis of the IWC retrieval is beyond the scope of this work. In the revised manuscript, we have added some text to discuss the retrieval uncertainty at typical values of $K_{dp}$ and $Z_{DR}$ in the HIWC regions (section 6, P21-22). In summary, standard deviation of IWC estimates at HIWC regions is about 0.6 $g/m^3$.

8. Fig. 12: I do not know the meaning of Fig. 12. Why can you see "the modified version of IWC is better replicated by a simple linear regression model" from this figure? "Modified IWC" is the just measured IWC scaled by Zdr. Why is this better replicated by a simple linear regression

model then measured IWC? In both Fig. 12a and Fig. 12b, it seems that there is ~50% variability in IWC at Kdp = 1. There is no significant difference between Fig. 12a and Fig. 12b. If you wanted to say that Eq. 13 can provide better correlations between IWC and polarimetric variables (Kdp and Zdr), this has been already shown by Rˆ2 values in Fig. 5.

The $Z_{dr}$ constraint is used to avoid large errors when $Z_{DR} \approx 1$. In general, Zdr values are not constant at different radar volumes; hence, $(1 - Z_{DR}^{-1})IWC$ is not a simply scaled version of IWC. We agree that the $R^2$ goodness of fit parameters in Fig. 5 already show that a linear regression fits $(1 - Z_{DR}^{-1})IWC$ better than IWC for the case of May 25. Figure 12 (now Fig. 14 in the revision) shows that the method also works well for other cases. In the revised manuscript, the sentence has been removed to avoid the confusion. We have also added estimated fitting coefficients for all the selected cases in Fig. 14 in Table 2 and discussed the variability in the coefficients in section 6.

9. "Modified IWC" is a confused phrase. In the manuscript, "modified IWC" represents the left hand side of Eq. 13. (1-ZDRˆ-1)*IWC. This is IWC scaled by [1-ZDRˆ-1], not actual IWC. Use an appropriate phrase through the text.

We completely agree with the reviewer. As it is also mentioned by another reviewer, "modified IWC" will confuse the readers. For a clear presentation, we have placed it with $(1 - Z_{DR}^{-1})IWC$ throughout the revised manuscript.

Minor comments

1. Did you do attenuation corrections for reflectivity and Zdr? Attenuation corrections may not be needed for pure ice precipitation, but it would be good to mention about this.

No, in this work, no attenuation correction was applied to reflectivity and $Z_{dr}$ as it pointed out by the referee that in ice precipitation and at close range, attenuation at X-band is negligible. Some text has been added to section 3.1 as suggested.

2. Did you do calibrate Zdr values for systematic offset?

Yes, we did.

3. What was the window size for calculating Kdp? What is the special resolution of Kdp?

The window size for calculating Kdp in our algorithm is 31 and the original Kdp range gate spacing (before decimation) is 30 m.

4. P. 10, lines 1-2 "The radar estimates. . .": This sentence is unclear. Please add detailed descriptions.

We thank the referee for this question. It should read "The IWC estimates from radar data …". The sentence has been changed in the revision.

5. P. 10, line 8 (2): This feature is unclear in Fig. 4. Scatter plots of MMD, RHOhv and Zdr should be helpful.

The data shown in Fig. 4 are measured time series data so we do not have scatter plots of the parameters. Between 11:16:-6 and 11:21:46 UTC, RHOhv decreased and Zdr increased at MMD peaks.

6. P. 10, line 9 and p. 14, line 2: How did you know the orientation of small crystals? Kdp should also be sensitive to larger crystals with aspect ratio /= 1. How did you know the particle aspect ratios?

We agree that, in general, the orientation of ice particles is unknown and saying $K_{dp}$ is only sensitive to small ice particle is not correct. The sentence in P10, L9 has been rewritten to remove this confusion. We have added a new figure (Fig. 6a in the revision) to show the typical values of the magnitude of $K_{dp}$ as a function of MMD for all the data used in this work. In general, $K_{dp}$ of large aggregates is smaller than that of small ice particles. The text in P14 has been revised to reflect this addition

7. P. 10, lines 10-11: In Fig. 5, it seems to me that the break point where Zdr started increasing and RHOhv deceased was at Kdp ~1.5 deg/km.

We agree with the referee. Corrected as suggested.

8. P. 10, line 20: I cannot see any blue line Fig. 5.

The line colors in Fig. 5 were changed from blue to black but not the text. We apologize for this confusion. The text has been corrected.

9. P. 10, lines 20-21: I cannot see measured IWC or modified IWC in Fig. 5.

In the revision, the measured IWC and $(1 - Z_{DR}^{-1})IWC$ (was "modified IWC") are in black.

10. Fig. 3b and Fig. 7b: Why is there a break line at 6-7 km range in reflectivity fields?

The break lines in those figures are locations of ground clutters which were filtered out. More information about the ground clutter contamination and filtering was given in section 3.1. In

the revised manuscript, we have added some text to the figure captions to explain for those lines.

11. Please give height information for Figs. 3, 7, 9, 10.

Aircraft height information has been added for the figures as suggested.

12. Please give information about height, range, and temperature for Figs. 4, 5, 6, 8, 11, 13.

We have added a sentence in section 3.1 to indicate that in this work, the radar profiles were extracted at 1000 m from the aircraft for all the study case. The information of aircraft altitude and static air temperature for Fig. 4, 5, and 6 (May 26 case) is given in section 5.1 and for Fig. 8, and 11 (May 23 case) is given in section 5.2. The intervals of aircraft altitude and static air temperature for Fig. 13 (all study cases) are given in the Fig. 13 caption.

13. Unit of Kdp should be degrees / km through the manuscript?

We thank the referee for this question. It was a mistake. Correction has been made.

14. Fig. 5: What do red lines represent?

They are the linear fitting curves to IWC and $(1 - Z_{DR}^{-1})IWC$ data. Last paragraph in P10 has been modified to include this information.

15. Fig. 5: I confused 'scatter plots' with red and black dots. I guess that 'scatter plots' here meant color shades representing frequency distributions. I recommend using 'frequency distribution' instead of 'scatter plot.'

We agree with the referee and thank for the comment. The caption has been changed as suggested.

16. Fig 5: missed (a), (b), and (c)

In the revised manuscript, all the figures have been checked and missing labels have been added.

17. P. 18 line 8: I cannot find ellipses in Fig. 11.

We apologize for missing ellipses in Fig. 11. They are now included in the revised figure.

18. Caption of Fig. 3: Delete "is"

The redundant word "is" has been removed from the Fig. 3's caption.

---

## Author Response (AR2)

**The authors well revised the manuscript taking into account all review's comments. I have minor comments below.**

1. Please provide brief case descriptions for: 1) the two cases in section 5.1 and section 5.2 and 2) each case in Table 2. The case description would be important to understand which types of clouds the technique can be applied to and which types of clouds are suitable for the technique. For example, what MCS type did each case represent; continental convective system or oceanic convective system? Are they multi cell convective system, leading stratiform type, or parallel stratiform type? What was the stage of these systems (e.g., developing/mature/dissipating)? Some readers may want to use the coefficients estimated in this study, because there are many cases that only radar data are available. For many observations, it is very difficult to obtain in-situ IWC and particle image data from aircraft rather than radar data. In that case, case descriptions are very useful to know which types of clouds are suitable for the technique and which coefficient values can be used.

Response:

We thank the reviewer for this suggestion. We have added information about the storm system as much as we know to sections 3 (L 4-8, P5), section 5.1 (L 18-19, P8) and 5.2 (L 9-10, P 14) in the revised manuscript. We've also added MCS type and size to each selected case in Table 2.

2. Were there differences in estimated coefficients or error values between convective regions and stratiform regions?

Response:

From our observation, IWC in stratiform regions are generally much lower than that in convective regions (about less than 0.5 g/m3 vs. 1-3 g/m3). Kdp shows a similar behavior (less than 0.4 deg/km vs. 1-3 deg/km). Hence, applying the linear regression to separate convective and stratiform regions will result different coefficients. The absolute values of errors are lower in the stratiform regions (due to smaller IWC) but the relative errors are in similar ranges.

3. I am not sure why the flow chart of Kdp estimation is more important than a flow chart of the IWC retrieval presented in this study (to me, a flow chart of the IWC retrieval is more useful). Can the Kdp estimation settings impact the IWC retrieval or error values? Can the Kdp estimation be a source of uncertainty in the coefficient estimates and IWC retrieval? When using different filtering size, can these values change?

Response:

The accuracy of Kdp estimate is very important in this work or, in general, any quantitative precipitation estimation which can be found in literature. Parameters used in Kdp estimation algorithm are normally tuned to work best for a given system. In our algorithm, Kpd estimates are not critically sensitive to the range filter size but large filter size will remove small scale feature in the Kdp field.

As suggested, we have added a flow chat of the IWC retrieval in section 2.2 of the revised manuscript.

4. Figures 6 and 12: Please add bias and RMS values for the Z-only estimate (green dashed line) for reference.

Response:

Added bias and rms values for IWC(Z) estimates as suggested.

5. P. 21, line 21: I think that "narrower" is not a good word for Fig. 14. Because the y axis is not the same scale in the two plots, it is very difficult to compare the variability. Can you use a different word or a different parameter comparing the two plots (e.g. normalized value); for example, standard deviation divided by maximum value (or divided by range of IWC ((1-Zdr^-1)*IWC))?

Response:

We completely agree with the referee. We have modified the Fig. 14 to include linear fits with coefficients computed from all data point (Table 2). Also, the y-axis are scaled to the maximum values of $IWC$ and $(1 - Z_{DR}^{-1})IWC$ for comparison. We found the legend in the figure was messed up and it is now fixed. The text in the manuscript has been also modified correspondingly to reflect the change.

In addition to the changes listed above, we've combined PIP images (Fig. 11) and 2DS images (Fig. 12) into a single figure (Fig. 11 in the revision) to improve the presentation. Last, major changes in this revision are highlighted in yellow.